METHODS AND RESOURCES

# Scalable eQTL mapping using single-nucleus RNA-sequencing of recombined gametes from a small number of individuals

Matthew T. Parker[1], Samija Amar[1], José A. Campoy[1¤a], Kristin Krause[1¤b], Sergio Tusso[2], Magdalena Marek[3], Bruno Huettel[3], Korbinian Schneeberger [1,2,4*]

1 Department of Chromosome Biology, Max Planck Institute for Plant Breeding Research, Cologne, Germany, 2 Faculty of Biology, Ludwig-Maximilians-Universität München, Planegg-Martinsried, Germany, 3 Max Planck Genome Center, Cologne, Germany, 4 Cluster of Excellence on Plant Sciences (CEPLAS), Heinrich-Heine University, Düsseldorf, Germany

¤a Current Address: Department of Pomology, Estación Experimental de Aula Dei (EEAD), Consejo Superior de Investigaciones Científicas, Zaragoza, Spain
¤b Current Address: Illumina Solutions Center Berlin, Berlin, Germany
* k.schneeberger@lmu.de

## Abstract

Phenotypic differences between individuals of a species are often caused by differences in gene expression, which are in turn caused by genetic variation. Expression quantitative trait locus (eQTL) analysis is a methodology by which we can identify such causal variants. Scaling eQTL analysis is costly due to the expense of generating mapping populations, and the collection of matched transcriptomic and genomic information. We developed a rapid eQTL analysis approach using single-cell/nucleus RNA sequencing of gametes from a small number of heterozygous individuals. Patterns of inherited polymorphisms are used to infer the recombinant genomes of thousands of individual gametes and identify how different haplotypes correlate with variation in gene expression. Applied to Arabidopsis pollen nuclei, our approach uncovers both *cis*- and *trans*-eQTLs, ultimately mapping variation in a master regulator of sperm cell development that affects the expression of hundreds of genes. This establishes snRNA-sequencing as a powerful, cost-effective method for the mapping of meiotic recombination, addressing the scalability challenges of eQTL analysis and enabling eQTL mapping in specific cell-types.

## Introduction

The proper regulation of gene expression is vital for the functioning of biological systems, serving to control the synthesis of proteins essential for cellular activities. Genetic variation between individuals often leads to measurable differences in gene expression, influencing an organism's phenotype at the molecular, cellular, and whole-organism level [1]. The discovery and understanding of these

which permits unrestricted use, distribution, and reproduction in any medium, provided the original author and source are credited.

**Data availability statement:** Illumina snRNA-seq data generated from pollen of five pooled F1 hybrids, and BGI snRNA-seq data generated from Col-0 × Db-1 hybrid pollen, are available from ENA project PRJEB77115. The Col-0 TAIR12 (Col-CC) assembly version 1 is available from NCBI GenBank accession GCA_028009825.1. The genome assemblies of the five parent 2 accessions are available from the MPI Edmond repository DOI:10.17617/3. AEOJBL. DAP-seq data for the E2FA transcription factor was downloaded from NCBI GEO accession GSE60143. Read counts, cluster labels and UMAP-projections for publicly available snRNA-seq data from developing pollen nuclei was downloaded from NCBI GEO accession GSE202422. Supplementary datasets for this manuscript are archived as Zenodo DOI 10.5281/zenodo.14864054. All pipelines, scripts and notebooks used to generate figures are available from GitHub at github. com/schneebergerlab/snrna_eqtl_mapping. git (copy archived as Zenodo DOI 10.5281/ zenodo.12633790).

**Funding:** This work was supported by the Deutsche Forschungsgemeinschaft (DFG) through grants EXC 2048/1-390686111 and Project-ID 456082119 – TRR 341/1 ("Plant Ecological Genetics") to KS, and the European Research Council (ERC) through grant 802629 ("INTERACT") to KS. The funders had no role in study design, data collection and analysis, decision to publish, or preparation of the manuscript.

**Competing interests:** The authors have declared that no competing interests exist.

**Abbreviations :** CIs, confidence intervals; CPV1, centromic pollen variant 1; eQTL, expression quantitative trait locus; LRT, log ratio test; PCA, principal components analysis; PSV1, pollen sperm variant 1; rHMM, rigid hidden Markov model; snRNA-seq, single cell or nucleus RNA-seq; UMIs, unique molecular identifiers.

expression-controlling variants is referred to as expression quantitative trait locus (eQTL) mapping [2]. eQTL mapping approaches have helped to identify many genetic factors that contribute to the diversity in traits observed among individuals [3].

eQTL mapping relies on the correlation between two sources of information: genetic variants within a population (the genotype), and gene expression profiles from the same population (the molecular phenotype) [2]. These measurements are conventionally collected using whole-genome resequencing and bulk RNA-sequencing (RNA-seq), respectively. Mapping populations can either be derived from sampling natural populations or experimentally generated through the controlled crossing of a limited number of founders [4]. Both approaches necessitate substantial time investment and pose scalability challenges. Larger sample sizes improve statistical power, but can be prohibitively expensive to obtain via conventional sequencing methods. Novel approaches are therefore required to advance the field of eQTL mapping.

Recent studies have applied single cell or nucleus RNA-seq (snRNA-seq) to human eQTL analysis, by pooling cell-lines from genetically distinct individuals to create heterogeneous "cell-villages" [5–8]. In snRNA-seq, the barcoding and sequencing of RNA molecules that derive from the same cell or nucleus is used to profile gene expression at the level of individual cells [9]. SNPs in sequenced reads are used to assign cells to their most likely donor of origin [10,11], and the known genotype of each donor is subsequently paired with gene expression information to perform eQTL mapping [5–8]. These methods have successfully identified many variants in *cis*-regulatory elements that result in eQTLs. A major benefit of snRNA-seq is the ability to detect cell-type specific eQTLs. However, all cells derived from the same individual share the same genotype, meaning that the resolution and power of the eQTL analysis is limited to the number of individuals used in the pool, rather than the number of cells in the dataset. The use of cell populations with greater genetic heterogeneity, such as recombinant haplotypes in gametes, or large populations of segregants [12,13], would therefore greatly improve the resolution of eQTL-mapping analysis with single-cell sequencing.

Here we present a method for rapid eQTL analysis using snRNA-seq of recombinant gametes from heterozygous individuals. We show that by using parental genotypes, variants identified from snRNA-seq reads can be used to infer the haplotypes of the individual gametes. The recombined haplotypes can be paired with gene expression estimates from the same nuclei to perform eQTL mapping. We apply this approach to explore the impact of natural variation on the haploid life cycle using mature Arabidopsis pollen nuclei. The vast majority of previous QTL and GWAS studies have been focused on the impact of genetic variation on the diploid life stage of organisms. In contrast to other developmentally complex eukaryotes, however, land plants have a haplodiplontic life cycle which involves mitotic cell divisions in not only their diploid, but also their haploid life phase [14]. Our analysis identifies numerous *cis*-eQTLs (located near to the gene they regulate), as well as many *trans*-eQTLs (located far from the gene they regulate) that map to the same hotspot on chromosome 1, implicating a potential master regulator of pollen sperm nucleus gene

expression. We conclude that snRNA-seq is a powerful approach for performing both meiotic recombination inference and eQTL analysis.

## Results

### Capture and sequencing of individual sperm cell nuclei from five Arabidopsis F1 hybrids

To determine whether individual gametes could be used for eQTL mapping, we generated a snRNA-seq dataset of nuclei from mature pollen collected from heterozygous Arabidopsis lines. We used five F1 hybrids produced by crossing Col-0 to the accessions Db-1, Kar-1, Ms-0, Rubezhnoe-1, and Tsu-0 (Fig 1A), for which high quality reference sequences were recently generated [15]. Mature pollen from these F1 hybrids was pooled, before being subjected to nuclei isolation using fluorescence activated cell-sorting (S1 Fig), and snRNA-seq using the 10x 3′ single-cell RNA-seq protocol (Fig 1A). After short read sequencing to saturation (S1 Table), reads were mapped to the reference sequence transformed with the major alleles of variants from the five parent 2 accessions [16]. Following barcode deduplication and nuclei whitelisting (see section "Cell barcode whitelisting"), we identified 1,394 high quality nuclei for analysis (Fig 1B).

Meiosis is not the end of pollen maturation: a post-meiotic developmental programme with two rounds of mitotic division and cell-type specification is required to generate mature trinuclear pollen, containing two sperm nuclei and a single vegetative nucleus, all three of which are genetically identical [14]. Previous analyses have shown that the two different classes of mature pollen nucleus have distinct transcriptomic programs that can be identified using specific marker genes [17–19]. In order to determine the nuclei types captured in our dataset, we performed principal components analysis (PCA) of nuclei gene expression (Fig 1B). This showed that the dataset captures two nuclei type clusters, which were separated primarily by the first principal component. Cluster 1 contained 1,305 nuclei, and these nuclei expressed marker genes such as *MGH3* and *PCR11* (Fig 1C) that are known to be specific to mature sperm nuclei [19–22]. Cluster 2 was much smaller, containing only 89 nuclei, and expressed marker genes such as *VCK* and *VGD1* (Fig 1C) that are known to be specific to mature vegetative nuclei [19,23,24]. As expected, the cluster representing vegetative nuclei had greater transcriptional diversity, with more expressed genes detected per nucleus after controlling for the number of unique molecular identifiers (UMIs; $p < 1 \times 10^{-16}$; S2 Fig) [19]. The lack of an approximately 2:1 ratio of sperm to vegetative nuclei in the dataset indicates that the isolation method used biases against vegetative nuclei, probably due to the uncondensed chromatin structure of these nuclei, which makes them larger, less dense and more fragile [25]. Alternatively, the gating strategy used for fluorescence activated cell sorting may have biased against the recovery of vegetative nuclei (S1 Fig). We conclude that our dataset successfully captures primarily mature sperm nuclei.

We next demultiplexed nuclei derived from each of the five F1 hybrid lines using parental genotypes and alleles present in the aligned reads [10,11]. Existing methods for discriminating cell genotypes from SNPs in snRNA-seq data expect clusters of cells to be genetically homogeneous and diploid [10,11], whereas recombinant pollen populations are genetically diverse and haploid. We therefore devised an alternative approach for clustering recombined haploid nuclei. Each nucleus contains genetic material inherited from Col-0 and one of the five other parental accessions. However, as many alleles are shared between the six parental accessions, we used fractional assignment and expectation maximization to create a probability score for each nucleus originating from each of the five possible F1 genotypes. We found that the majority of nuclei could be assigned to one F1 genotype with high confidence using this approach (Fig 1D-E). The remaining nuclei that are not confidently assigned are likely to be doublets (a form of technical error where two different nuclei have been assigned the same cell barcode) or other artefacts containing multiple nuclei or high levels of ambient RNA from different F1 accessions. The genotyping score was therefore used as a metric for whitelisting of high-quality nuclei. The median assignment probabilities for Col-0 × Db-1 and Col-0 × Tsu-0 were marginally higher than for the other three F1 genotypes (Fig 1E), likely due to the greater number of uniquely identifying SNPs in these two lines (S3 Fig). There was variation in the number of recovered nuclei for the five genotypes, with approximately three times as many

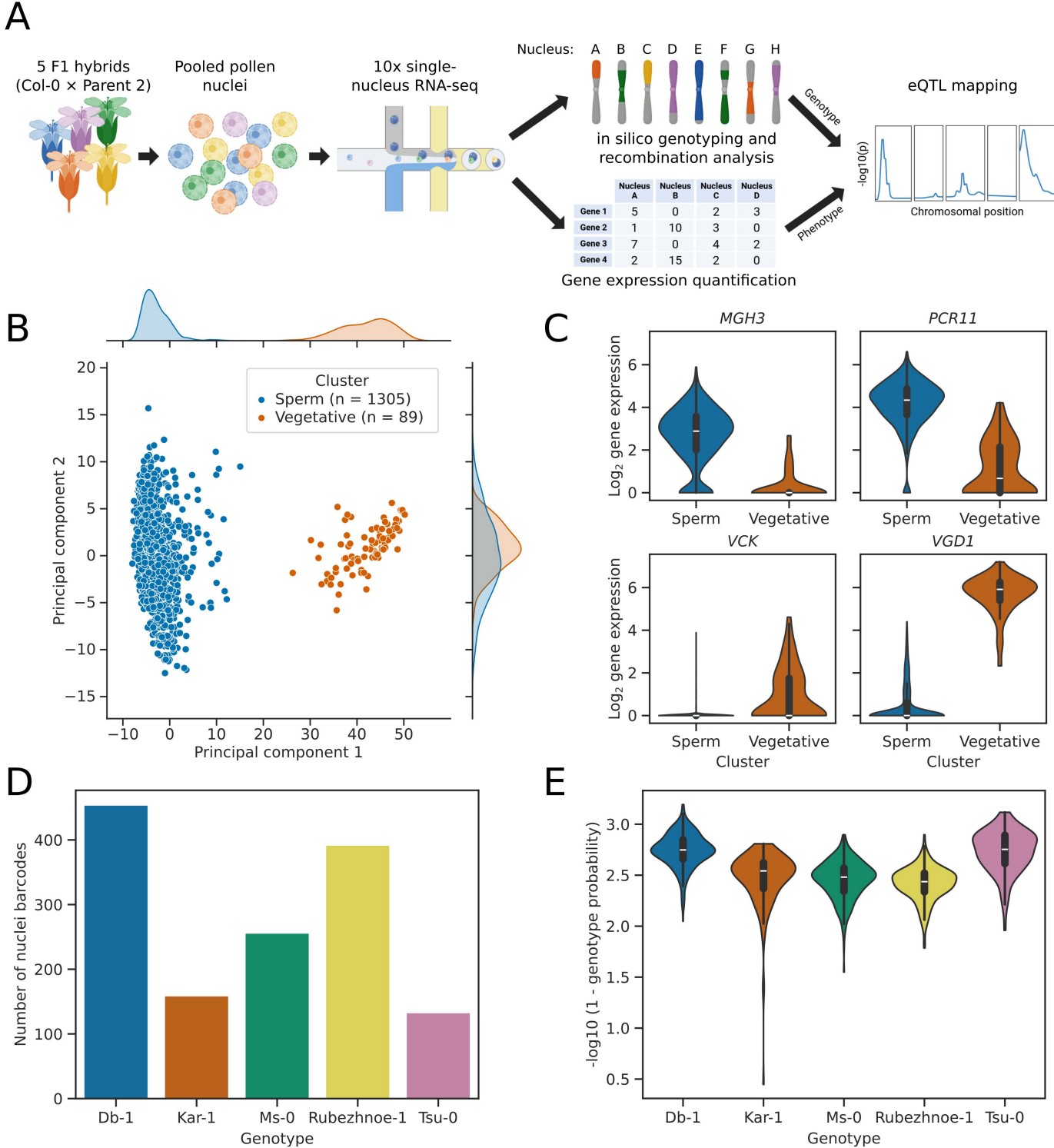

**Fig 1. snRNA-seq of mature pollen nuclei from five different Arabidopsis F1 hybrids. (A)** Diagram of snRNA-seq analysis pipeline. Heterozygous Arabidopsis F1 plants were generated by crossing Col-0 to five different accessions. Pollen was collected from these five F1 plants and pooled, before nuclei isolation, barcoding, and cDNA synthesis using the 10x snRNA-seq protocol. After sequencing, nuclei from the five genotypes were demultiplexed, and meiotic recombination events were predicted. Gene expression and recombination patterns were then compared to identify eQTLs. Figure

created using BioRender. **(B)** Scatter plot with marginal kernel density estimates, showing the first two principal components of the expression data for 1,394 pollen nuclei. Nuclei form two unequally sized clusters, separated by the first principal component. **(C)** Violin plots showing the expression of known marker genes for sperm and vegetative nuclei. Expression of genes *MGH3* and *PCR11* indicates that the blue cluster corresponds to sperm nuclei, whilst expression of the marker genes *VGD1* and *VCK* indicate that the orange cluster corresponds to vegetative nuclei. **(D)** Bar plot showing the number of high-quality nuclei in the dataset predicted to originate from each of the five different F1 hybrids. **(E)** Violin plot showing the genotyping score for high quality nuclei from each of the five different F1 hybrids. The data underlying this figure can be found in dataset 1 at https://doi.org/10.5281/zenodo.14864053.

Col-0 × Db-1 nuclei as Col-0 × Tsu-0, for example (Fig 1D). In summary, we were able to recover pollen sperm nuclei derived from all F1 lines in the pool and confidently genotype the vast majority of sperm cell nuclei.

### Identification of meiotic recombination in individual pollen nuclei using RNA sequencing

Our next goal was to use markers from each nucleus to identity meiotic crossover events. Variants that distinguished each of the five parental accessions from Col-0 were identified and used to locate marker reads for each genotyped nucleus. We subdivided the chromosomes into 25 kb bins, and Col-0 and parent 2 marker counts for each bin were calculated. High quality nuclei had a median of 481 marker reads (Fig 2A), with a median distance of 350 kb between non-empty, informative marker bins (S4A Fig). Marker reads were not uniformly distributed across the genome, with a lack of marker reads in regions with low gene density and in centromeres (Fig 2B), since these regions also exhibit low gene expression.

We adapted the recently described rigid Hidden Markov model (rHMM) approach for recombination mapping, for use in haploid nuclei [26]. The rHMM method uses sparsely connected states, meaning that any path through the model passes through multiple states representing the same haplotype. The benefit of this architecture is that it can be used to enforce a minimum distance separating switches between haplotypes, mimicking the natural process of crossover interference that inhibits crossovers from occurring close to one another in sequence space [27]. To simulate observed patterns of crossover interference in Arabidopsis a rigidity of 100 bins was used, with each bin representing 25 kb of genomic sequence, meaning that the minimum allowable distance between adjacent crossovers in any path through the rHMM was 2.5 Mb [27]. For each nucleus, the probability that each chromosomal bin was inherited from either Col-0 or parent 2 was estimated using the Forward–Backward algorithm, and the probability of a crossover occurring between each pair of adjacent bins was calculated. This identified a mean of 4.59 crossovers per nucleus, slightly below the 5.4 crossovers per male meiosis reported for Col-0 × Ler-0 hybrids [28]. This may be due to genotype-specific differences in recombination rate, or to the sparser nature of the markers identifiable in snRNA-seq, which could lead to some crossovers being missed, particularly at chromosome ends. An example of the marker distribution and resulting haplotype predictions for a single nucleus are shown in Fig 2C.

Using the haplotype probabilities from the rHMM, we measured the resolution with which crossovers could be located using 95% confidence intervals (CIs). These CIs showed a bimodal distribution depending on their proximity to the centromere – crossovers that were predicted to occur in the chromosome arms had a median 95% CI length of 1.1 Mb, whilst crossovers with CIs that overlapped the centromere had a median 95% CI length of 8.2 Mb (S4B Fig).

There was variation in the number of detected crossovers between genotypes – Col-0 × Db-1 nuclei had an average of 5.21 crossovers per nucleus, whereas in Col-0 × Rubezhnoe-1 nuclei we detected an average of only 4.39 (Fig 2D). Col-0 × Tsu-0 F1 hybrids, which were previously shown by reporter-based approaches to have a lower than average recombination rate across five different chromosomal intervals [29], behaved similarly to Col-0 × Rubezhnoe-1, with 4.28 crossovers per nucleus. We used the estimated mosaic of parental haplotypes (S5 Fig) to estimate crossover position distributions for each hybrid (S6 Fig). Crossover patterns were similar to previously described patterns identified in Arabidopsis pollen, with no recombination in centromeres, and peaks of recombination in pericentromeric regions and towards chromosome ends [28,30]. We conclude that variants from snRNA-seq are sufficient to genotype recombinant pollen genomes.

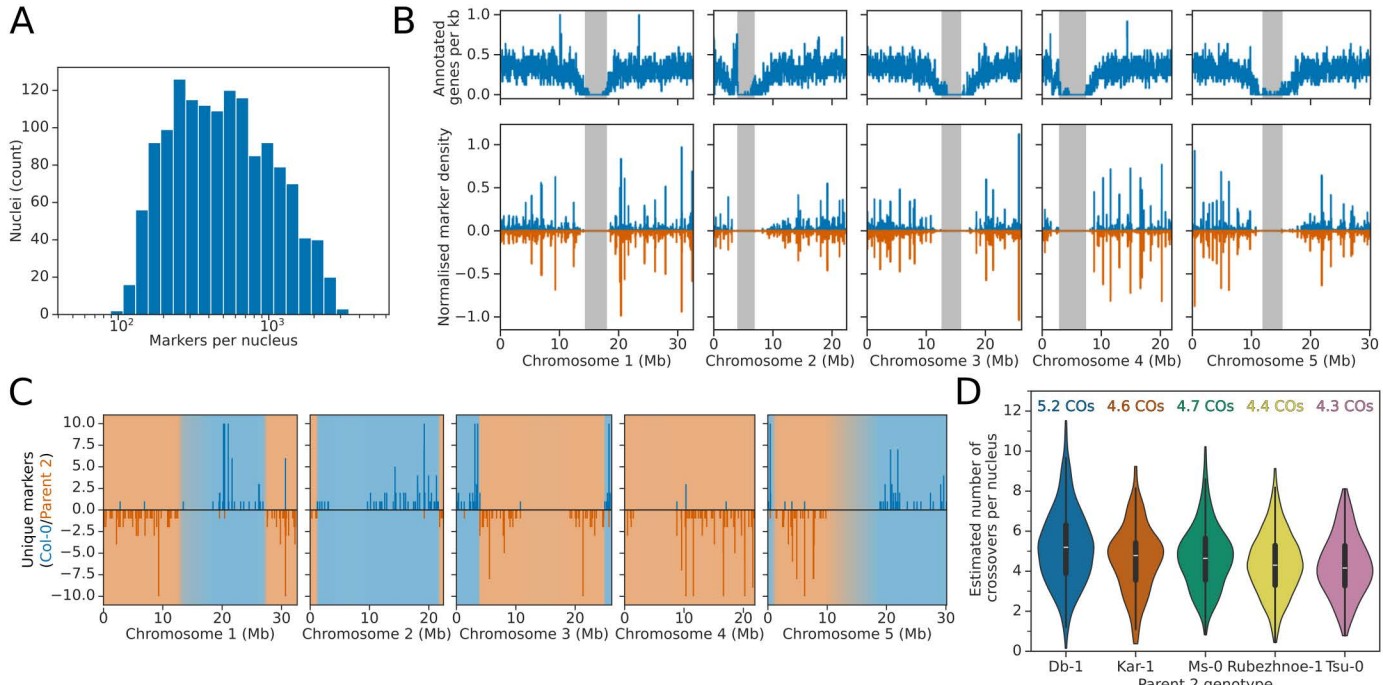

**Fig 2. Identification of meiotic recombination patterns from snRNA-seq data. (A)** Log$_{10}$-scale histogram showing the effective number of unique markers identified for each high-quality nucleus in the snRNA-seq dataset. **(B)** Histogram showing the distribution of genes (top panels) and markers for Col-0 and parent 2 (bottom panels, Col-0 in blue, parent 2 in orange) across chromosomes. Centromere locations are shown as grey bands. **(C)** Genome-wide plot showing marker positions and predicted meiotic recombination events for an example nucleus. Markers which support the Col-0 genome are shown in blue above the axis, whilst markers supporting the Parent 2 genome are shown in orange, below the axis. Regions predicted by the rHMM to originate from the Col-0 or Parent 2 are shown with blue or orange background shading, respectively. **(D)** Violin plots showing the predicted number of recombination events per nucleus for the five different F1 hybrids. Mean crossover numbers for the five different F1 hybrids are shown above each violin. The data underlying this figure can be found in dataset 2 at https://doi.org/10.5281/zenodo.14864053.

### Expression quantitative trait locus mapping using single nucleus genotypes

A major advantage of determining genotypes from snRNA-seq, rather than DNA-sequencing, is that expression profiles are collected at the same time as genotyping information. These two sources of information can then be combined for eQTL mapping [12]. To efficiently perform this analysis, we used the estimated haplotypes in 25 kb bins to perform linear modelling. Rather than discretising haplotype predictions from the rHMM method, we used probabilistic estimates directly to propagate uncertainty from haplotype predictions to the eQTL mapping stage. For all pairwise combinations of haplotype bin and expressed gene, we used this information to fit multivariate linear models to determine whether predicted variation in the haplotype could explain the observed variation in gene expression. To control for cell type and technical variation, we included principal components derived from the expression analysis as covariates [31]. We also included parental genotypes as a covariate to control for population structure in the dataset. This method identified 293 genes that had a difference in gene expression associated with the assortment of at least one haplotype bin, at an FDR threshold of 5% (Fig 3A).

For genes with differential expression correlated with the inheritance of parental haplotypes, we used peak calling to identify the most significantly associated haplotype bin. This resulted in the identification of 326 eQTL positions (Fig 3A) – an average of 0.14 eQTLs per tested gene. Of the identified eQTLs, 67.2% were physically located within 2 Mb of the gene whose expression they correlated with, indicating that the majority are caused by *cis*-acting genetic variation

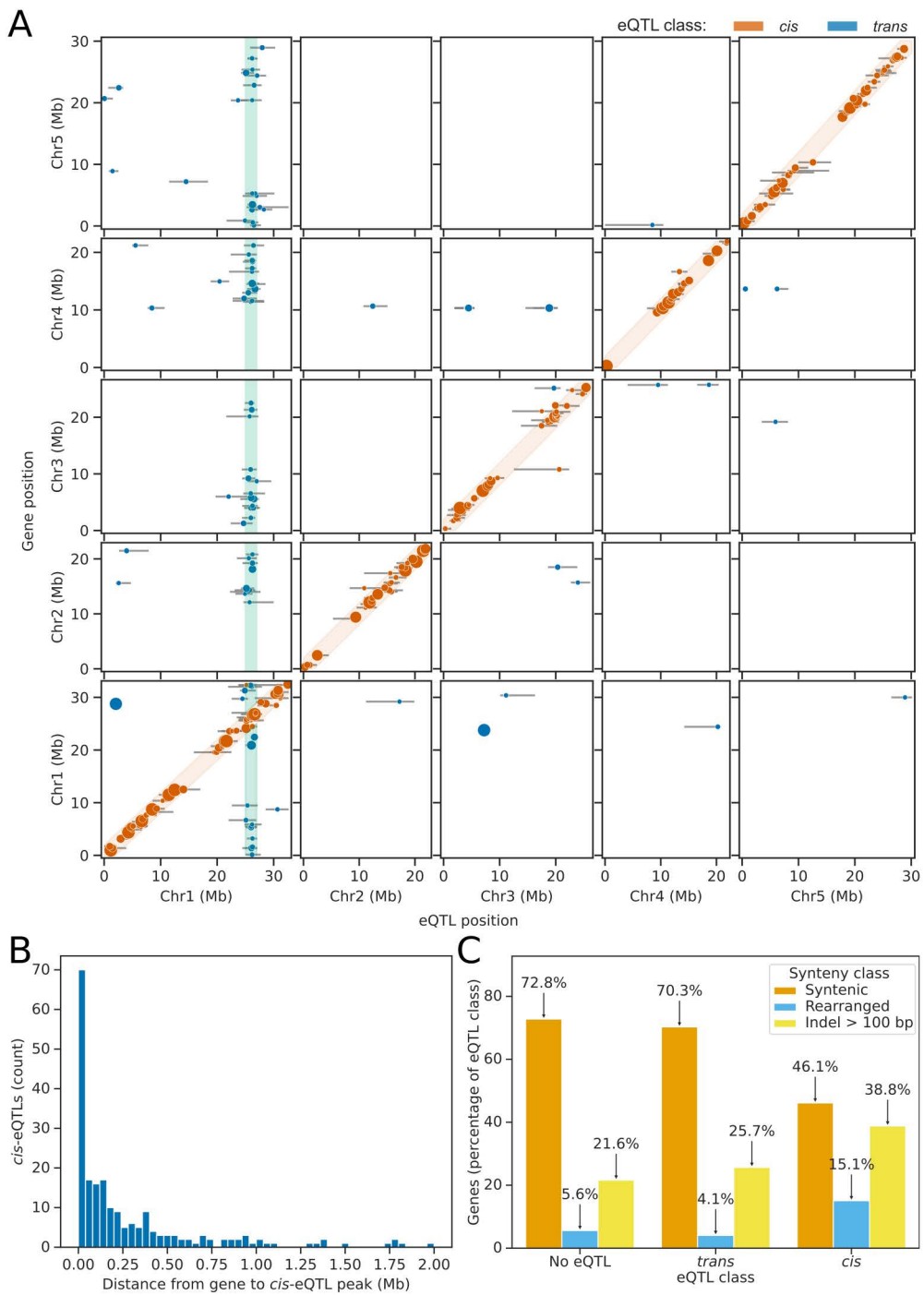

**Fig 3. eQTL mapping identifies *cis*- and *trans*-eQTLs in Arabidopsis pollen nuclei. (A)** Genome-wide map of eQTL positions (*x* axis) shown relative to the genomic position of the gene whose expression was tested (*y* axis). Error bars represent the confidence intervals for each eQTL determined using the 1.5 LOD-drop method, and the size of the points is directly proportional to the LOD score of the eQTL. The orange shaded diagonal indicates the region within which eQTLs were considered to be *cis*-eQTLs. The green shaded region indicates the hotspot of *trans*-eQTLs referred to as *POLLEN SPERM VARIANT 1 (PSV1)*. **(B)** Histogram showing the distance between the genomic location of genes with *cis*-eQTLs affecting their gene expression and the mapped location of the *cis*-eQTL. **(C)** Barplots showing that genes with *cis*-eQTLs are enriched for rearrangements and large insertions or deletions, compared to genes expressed in pollen which do not have *cis*-eQTLs. The data underlying this figure can be found in dataset 3 at https://doi.org/10.5281/zenodo.14864053.

(*cis*-eQTLs). The median distance between the mapped location of *cis*-eQTLs and the gene whose expression they correlated with was 141 kb (Fig 3B).

Some *cis*-eQTLs may be caused by presence–absence variation or other larger rearrangements of the genes. To determine whether genes with a *cis*-eQTL peak were associated with larger rearrangements expected to affect expression, we aligned the chromosome scale assemblies of the parental genomes and performed synteny analysis. Of the 2,108 genes tested that did not have an eQTL peak, 72.8% were located in regions which were syntenic with Col-0, without large indels, in all five parental accessions (Fig 3C). Similarly, of the 74 genes with *trans*-eQTL peaks, 70.3% were located in syntenic regions ($\chi^2$ $p$ = 0.64; Fig 3B). In comparison, of the 228 genes with a *cis*-eQTL peak, only 46.1% were in regions fully syntenic with Col-0 ($\chi^2$ $p$ = 1.7 × 10$^{-16}$; Fig 3C). This indicates that genes with detectable *cis*-eQTLs are enriched for rearrangements. For example, we found that only nuclei inheriting the Col-0 or Kar-1 alleles of the gene AT1G31990, which encodes a sperm-cell specific protein of unknown function [19,32], express the AT1G31990 gene (Fig 4A-B). Synteny analysis of the 6 parental genomes [15] indicates that this is due to deletion of the region containing AT1G31990 in these accessions, which are not present in Col-0 or Kar-1 (Fig 4C).

*Cis*-eQTLs were also identified that could be explained by insertions of transposable elements. An example of this was found at AT3G12510, which encodes a MADS-box transcription factor (S7A Fig). Nuclei that inherit the Kar-1 allele of AT3G12510 do not express the AT3G12510 gene (S7B Fig). This is likely due to the insertion of a LINE/L1 retrotransposon immediately upstream of the AT3G12510 gene, which is present only in Kar-1 (S7C Fig), and likely results in the silencing of the gene.

Another *cis*-eQTL appears to control the expression of the gene *CTF7* (AT4G31400), which encodes an acetyltransferase that is required for sister chromatid cohesion during both mitosis and meiosis (Fig 4D) [33]. Pollen nuclei that inherit the Col-0, Ms-0, Rubezhnoe-1 or Kar-1 alleles of *CTF7* express *CTF7* mRNA at levels mostly below the detection threshold of the snRNA-seq experiment (Fig 4E). In contrast, nuclei that inherit the Tsu-0 or Db-1 alleles of *CTF7* were much more likely to have detectable *CTF7* expression (Fig 4E). Sequence analysis of the Col-0 *CTF7* gene indicates that the promoter region contains two copies of a 24 nt tandem repeat (Fig 4F). This tandem repeat region occurs in the open chromatin of the *CTF7* promoter and contains two copies of the binding motif GGCGCCA, which is directly bound by the transcription factor E2FA to regulate *CTF7* expression [34,35]. The Tsu-0 and Db-1 alleles of *CTF7*, however, contain three copies of this tandem repeat, thereby likely introducing an extra E2FA binding site (Fig 4F). This extra transcription factor binding site may explain the increased expression of *CTF7* in pollen nuclei inheriting the Tsu-0 and Db-1 *CTF7* alleles. We conclude that eQTL mapping from pollen snRNA-seq data can identify genuine and explainable interactions between genetic and transcriptomic variation.

## Pollen eQTL mapping identifies a locus on Chr1 associated with the expression of multiple sperm-nucleus-specific genes

Of the 107 *trans*-eQTL peaks, a remarkable 65.4% were located at the same *trans*-eQTL hotspot identified at approximately 26 Mb on Chromosome 1 (Fig 5A). This suggests that there is natural variation at this locus in a gene which acts as a general regulator of pollen gene expression. We named this locus *POLLEN SPERM VARIANT 1 (PSV1)*. For example, we identified variation in the expression of the mRNA poly(A)-binding protein coding gene *PAB7*, that depends on the genotype of the *PSV1* locus (Fig 5B). Nuclei that inherit the Kar-1, Ms-0 or Tsu-0 allele of *PSV1* show similar expression of *PAB7* to sibling nuclei that inherit the Col-0 allele (Fig 5C). In contrast, nuclei that inherit the Rubezhnoe-1 or Db-1 allele of *PSV1* have higher expression of *PAB7* than sibling nuclei that inherit the Col-0 allele (Fig 5C). A similar *trans*-eQTL mapping to the *PSV1* locus was also identified for the related poly(A) binding protein *PAB6* (S8A Fig). Despite significant divergence between the *PAB7* and *PAB6* genes [36], the expression of *PAB6* is affected similarly to *PAB7*, with increased expression observed when the Db-1 or Rubezhnoe-1 allele of *PSV1* is inherited (S8B Fig). The expression of *PAB6* and *PAB7* is highly specific to mature tricellular pollen [37], and analysis of snRNA-seq data from multiple post-meiotic stages

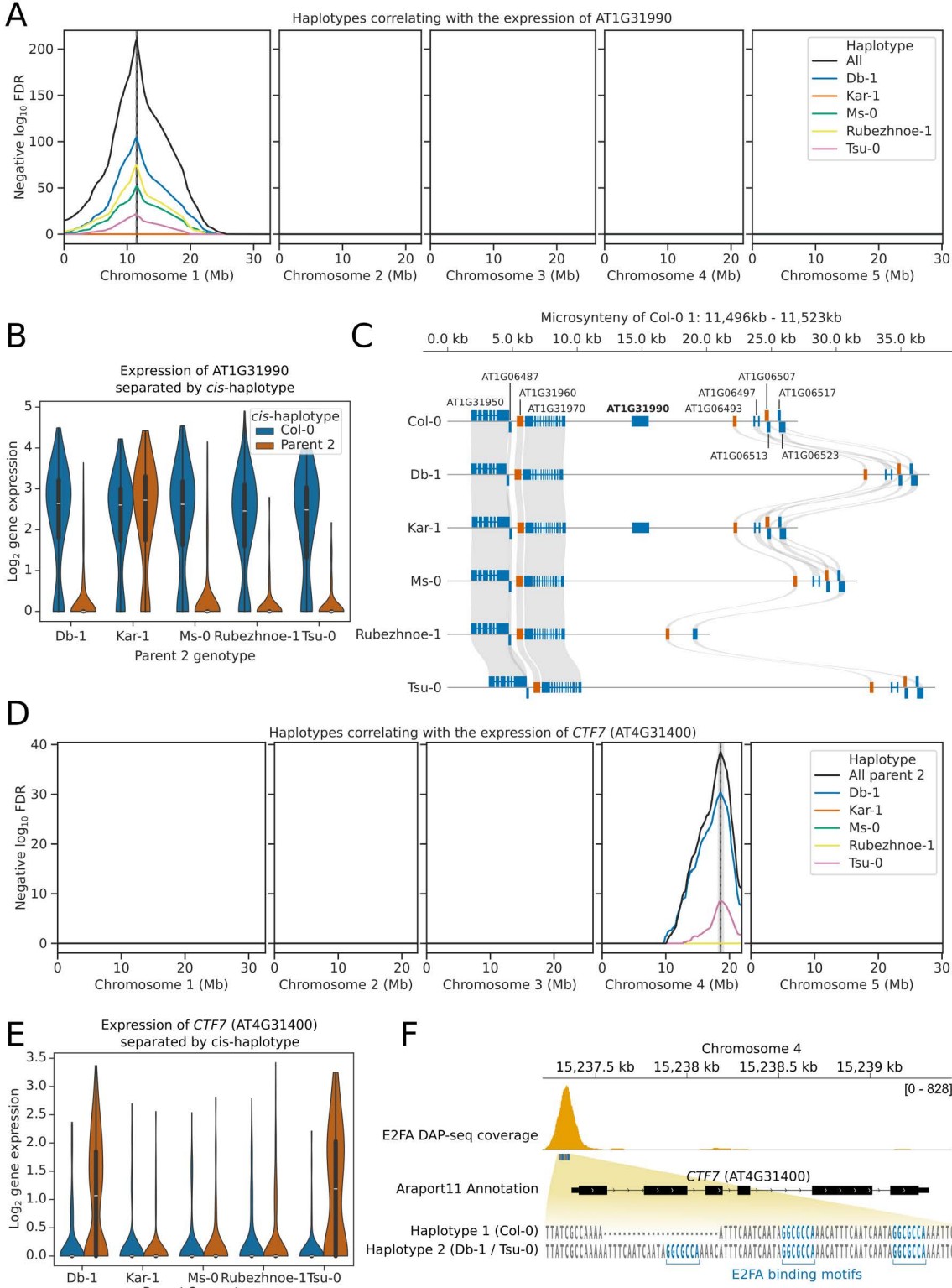

**Fig 4. Pollen snRNA-seq identifies *cis*-eQTLs.** (A) eQTL plot showing the haplotypes whose inheritance correlates with the expression of the gene AT1G31990. eQTL peaks are shown as vertical dashed lines with 1.5 LOD drop confidence intervals shown as grey shaded regions. The location of the AT1G31990 gene is shown as a solid vertical black line. The black line labelled "All" shows the FDR calculated from the log ratio test of all 5 parent 2 haplotypes compared to Col-0. **(B)** Violinplot showing the gene expression of AT1G31990 in nuclei separated by the *cis*-haplotype (i.e., the haplotype at

AT1G31990). Nuclei that inherit the Db-1, Ms-0, Rubezhnoe-1, or Tsu-0 haplotype of AT1G31990 have significantly reduced expression of AT1G31990, compared to sister nuclei that inherit the Col-0 haplotype. **(C)** Synteny plot showing the conservation of the AT1G31990 locus and neighbouring loci in the 6 accessions used in this study. AT1G31990 is conserved only in Col-0 and Kar-1. **(D)** eQTL plot showing the haplotypes whose inheritance correlates with the expression of the gene *CTF7* (AT4G31400). eQTL peaks are shown as vertical dashed lines with 1.5 LOD drop confidence intervals shown as grey shaded regions. The location of the *CTF7* gene is shown as a solid vertical black line. The black line labelled "All" shows the FDR calculated from the log ratio test of all 5 parent 2 haplotypes compared to Col-0. **(E)** Violinplot showing the gene expression of *CTF7* in nuclei separated by the *cis*-haplotype (i.e., the haplotype at *CTF7*). Nuclei that inherit the Db-1, or Tsu-0 haplotype of *CTF7* have significantly increased expression of *CTF7*, compared to sister nuclei that inherit the Col-0 haplotype. **(F)** Gene track showing promoter haplotypes of the *CTF7* locus between Col-0, Db-1 and Tsu-0. *CTF7* is a known target of the E2F family of transcription factors that bind to a GGCGCCA motif in the *CTF7* promoter (DAP-seq binding peak of E2FA from O'Malley and colleagues 2016 is shown in orange). The Db-1/Tsu-0 haplotype of *CTF7* contains an extra copy of this binding motif, which may affect E2F recruitment and explain *CTF7* expression changes. The data underlying this figure can be found in datasets 1, 2 and 3 at https://doi.org/10.5281/zenodo.14864053.

of developing pollen indicated that they are specific to the sperm nucleus, with no expression in vegetative nuclei or in generative nuclei before pollen mitosis II (Fig 5D, S8C Fig) [19,32]. More broadly, we found that of the 70 genes with a *trans*-eQTL peak at *PSV1*, 97.1% had higher expression in sperm nuclei than in vegetative nuclei, compared to only 65.8% of all tested genes (S9A Fig). Furthermore, analysis of snRNA-seq data from developing pollen indicated that genes with *PSV1 trans*-eQTLs are highly restricted to the sperm nuclei, with limited or no expression in mono- or bicellular pollen nuclei (S9B Fig) [19]. There was also significant overlap between the genes with a *PSV1 trans*-eQTL that was significant specifically in Db-1 and Rubezhnoe-1 (Jaccard index: 0.16, hypergeometric $p = 8.7 \times 10^{-9}$, S9C Fig), suggesting that the underlying locus is the same in both accessions. We conclude that *PSV1* encodes a general regulator controlling the expression of many genes that are expressed primarily in pollen sperm nuclei.

### *Trans*-eQTL hotspots identified by snRNA-seq are replicable across datasets

To replicate our findings, and more finely map the position of the *PSV1* locus, we performed a second snRNA-seq experiment using only nuclei isolated from Col-0 × Db-1 hybrid pollen, again using fluorescence activating cell-sorting and the 10x Chromium platform (S10 Fig). This experiment yielded a total of 7,458 high quality barcodes, which were predicted to represent 6,564 (88.0%) sperm nuclei and 894 (12.0%) vegetative nuclei (S11 Fig). We believe that the marginally improved recovery of vegetative nuclei in this second dataset is the result of more gentle handling and the omission of some centrifugation steps, which may have caused greater lysis of the fragile vegetative nuclei in the first experiment [25].

Haplotyping and eQTL mapping was performed as for the first dataset, using the expression of 2,622 genes as phenotypes. Due to the single parental genotype and increased number of nuclei in the second dataset, there was much greater statistical power, resulting in the recovery of a much larger number of eQTLs. In total, 1,227 genes (46.8%) had at least one eQTL peak, with an average of 0.63 eQTLs per tested gene, and a total of 1,639 eQTL peaks identified overall (Fig 6B). Of these eQTLs, 577 (35.2%) were *cis*- acting. *Cis*-eQTLs could be mapped with remarkable accuracy, with a median distance of 155 kb between the eQTL peak and position of the gene (S12A Fig). Furthermore, 27.7% of *cis*-QTLs were mapped to the exact 25 kb haplotype bin containing the gene whose expression they correlated with. Of the 79 Db-1 specific *cis*-eQTLs recovered in the first dataset, 88.6% were also identified in the second dataset, demonstrating the reproducibility of the results (S12B Fig).

Compared to the first dataset, a higher proportion of *trans*-acting eQTLs were identified. In total, 1,070 (64.8%) eQTLs acted in *trans.* Of these, 560 (54.0%) overlapped with the *PSV1* locus at 26 Mb on Chromosome 1 (Fig 6A). These included very strong *trans*-eQTL hits in genes that we also identified in the first dataset, such as *PAB6*, *PAB7*, *UBC20* and *MAPKKK20* (Fig 6B). In total, of the 24 PSV1 *trans*-eQTLs identified in the first dataset that were significant in Db-1 compared to Col-0, 83.3% were also identified in the second dataset (Fig 6C). Furthermore, 100% of the 18 Rubezhnoe-1 *trans*-eQTLs at PSV1 were also recovered, indicating that the effect of the causal locus at *PSV1* in Db-1 and Rubezhnoe-1 is likely to be the same, and could result from a shared variant or haplotype.

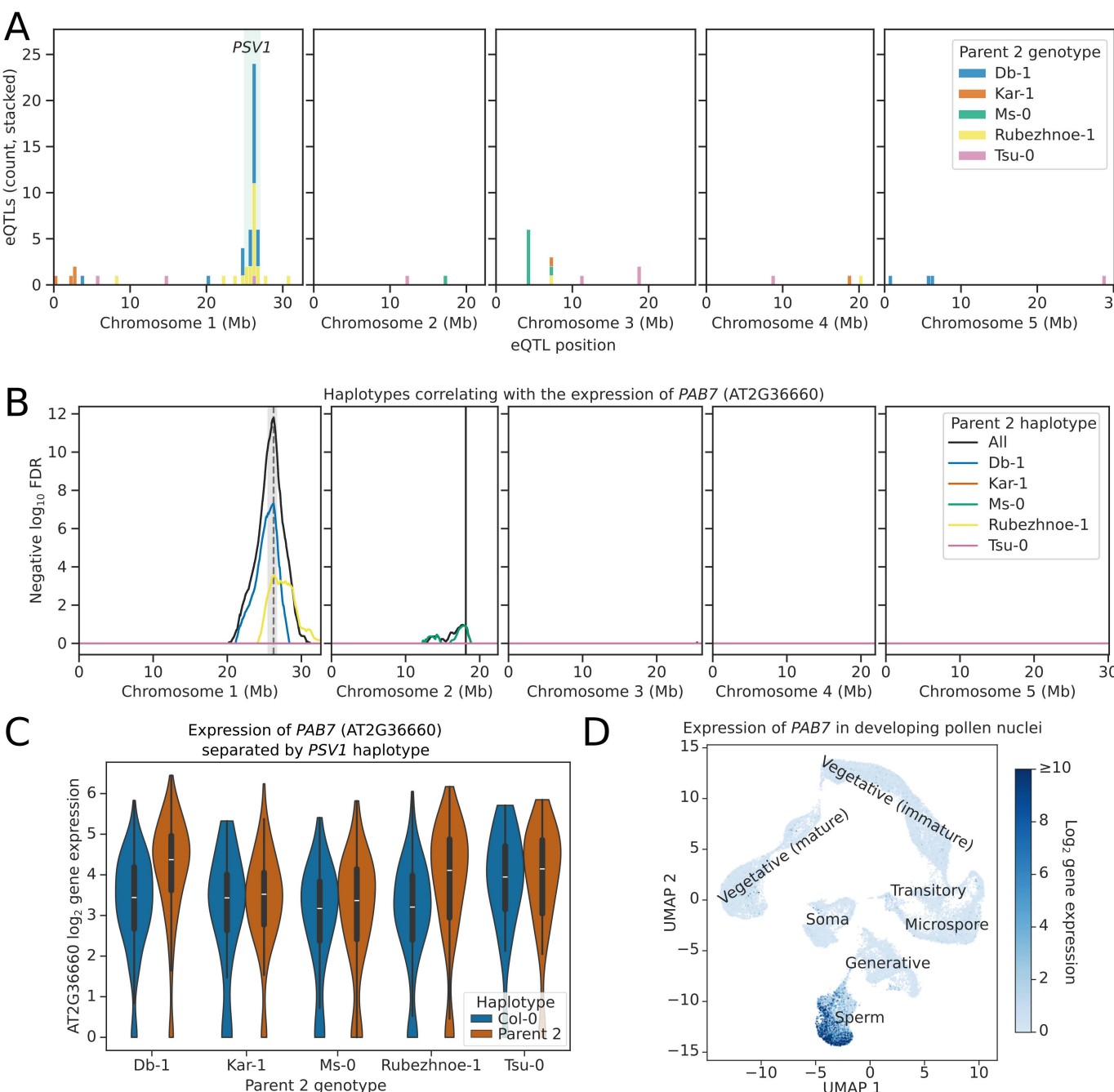

**Fig 5. Pollen snRNA-seq identifies a *trans*-eQTL hotspot on Chromosome 1. (A)** Histogram showing the distribution of parent 2 specific *trans*-eQTL hits across the Col-0 genome. *trans*-eQTL hits were filtered for those that were significant at the 5% threshold in each parent 2 contrast. A *trans*-eQTL hotspot was identified at Chr1:26 Mb, which was named *PSV1* (green vertical shaded region). **(B)** eQTL plot showing the haplotypes whose inheritance correlates with the expression of the gene *PAB7* (AT2G36660). eQTL peaks are shown as vertical dashed lines with 1.5 LOD drop confidence intervals shown as grey shaded regions. The location of the *PAB7* gene is shown as a solid vertical black line. The black line labelled "All" shows the FDR calculated from the log ratio test of all 5 parent 2 haplotypes compared to Col-0. **(C)** Violinplot showing the gene expression of *PAB7* in nuclei separated by the *PSV1*-haplotype. Nuclei that inherit the Db-1 or Rubezhnoe-1 haplotype of *PSV1* have significantly higher expression of *PAB7*, compared to sister nuclei that inherit the Col-0 haplotype. **(D)** UMAP projection from Ichino and colleagues 2022, showing the expression of *PAB7* throughout the developmental stages of the pollen. *PAB7* is only expressed in the sperm nucleus cluster, and is absent from microspore, generative and vegetative nuclei, as well as from the soma. The data underlying this figure can be found in datasets 1, 2 and 3 at https://doi.org/10.5281/zenodo.14864053.

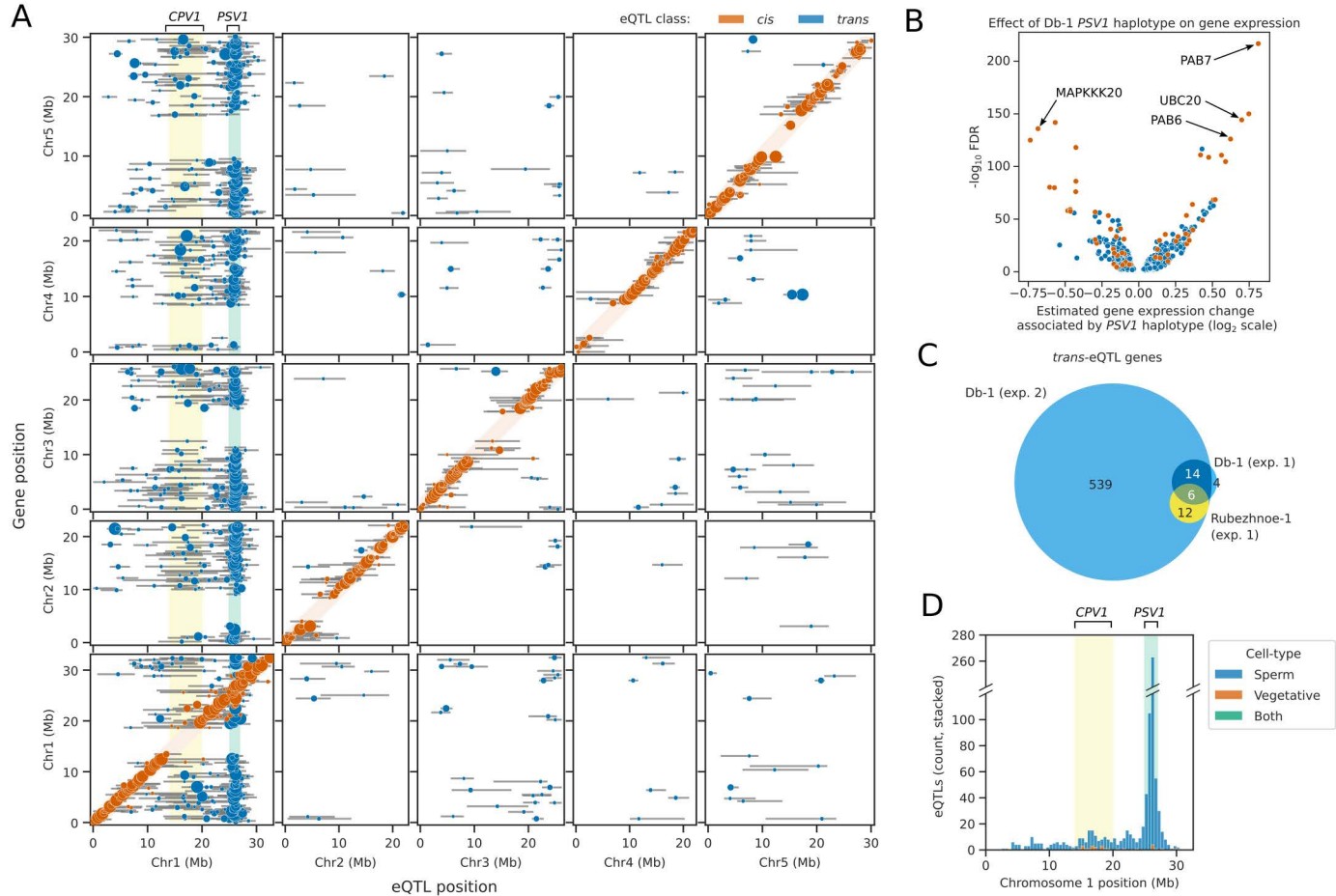

**Fig 6. Reproducibility of the PSV1 trans-eQTL hotspot across experiments.** (A) Genome-wide map of eQTL positions (x axis) shown relative to the genomic position of the gene whose expression was tested (y axis). Error bars represent the confidence intervals for each eQTL determined using the 1.5 LOD-drop method, and the size of the points is directly proportional to the LOD score of the eQTL. The orange shaded diagonal indicates the region within which eQTLs were considered to be *cis*-eQTLs. The green shaded region indicates the PSV1 hotspot of *trans*-eQTLs, and the yellow shaded region indicates the CPV1 *trans*-eQTL hotspot. (B) Volcano plot showing the modelled effect of the Db-1 PSV1 haplotype on the expression of genes with a *trans*-eQTL peak at PSV1. Blue dots represent genes with novel PSV1 trans-eQTLs identified only in the second dataset, whilst orange dots represent genes which also had PSV1 trans-eQTLs identified in the first dataset. (C) Venn-diagram showing the overlap of PSV1 *trans*-eQTL genes identified in Db-1 and Rubezhnoe-1 compared to Col-0 in the first dataset, with the PSV1 trans-eQTL genes identified in Db-1 compared to Col-0 in the second dataset. (D) Histogram showing the distribution of cell/nucleus-type specific *trans*-eQTLs identified on Chromosome 1 in sperm and vegetative nuclei. A broken axis is used to allow inspection of the CPV1 locus, which is broader and has fewer significant *trans*-eQTLs than the PSV1 locus. The data underlying this figure can be found in datasets 4 and 5 at https://doi.org/10.5281/zenodo.14864053.

A second putative *trans*-eQTL hotspot was also identified on Chromosome 1 in the centromere in the region between approximately 14 and 20 Mb, which accounted for a further 213 (20.1%) of *trans*-eQTLs. This QTL hotspot was much broader than that identified for *PSV1*, reflecting reduced meiotic recombination frequencies across centromeric regions (Fig 6A, 6D). We therefore named this locus *CENTROMIC POLLEN VARIANT 1 (CPV1)*. In total, 85.1% of all *trans*-eQTLs mapped to Chromosome 1. Assuming that false positives are distributed randomly across all chromosomes, this would indicate a low rate of false discovery. We conclude that the identification of the *PSV1* locus is replicable across datasets.

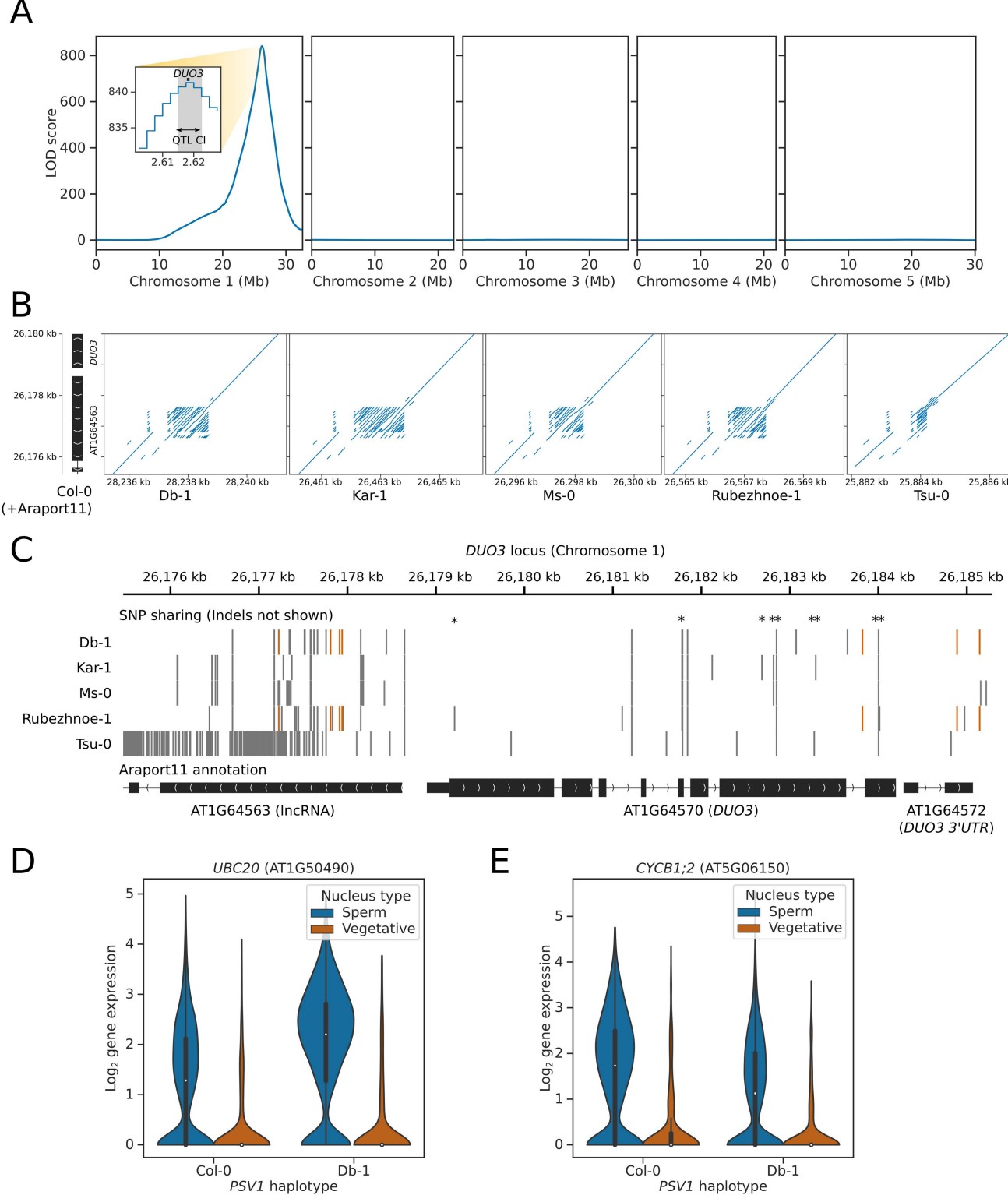

**Fig 7. Fine mapping of the *PSV1* locus implicates the mitotic cell cycle regulator *DUO3*. (A)** QTL plot showing the haplotypes whose inheritance correlates with the first principal component of the expression of genes which have a *PSV1 trans*-eQTL peak. Inset shows the fine-mapping of the *PSV1* locus. Confidence intervals for the QTL locus (QTL CI) were identified using the 1.5 LOD drop method (grey shaded area). The candidate gene *DUO3* is

located at the center of the *PSV1* interval (*DUO3* location shown in black). **(B)** Dotplots showing the structure of the *DUO3* promoter in the five parent 2 accessions compared to Col-0. The promoter region contains a hyper allelic tandem repeat. **(C)** Gene track showing the SNP sharing patterns between the five parent 2 accessions at *DUO3* and the upstream gene/promoter region AT1G46563. Missense variants are denoted with asterisks. SNPs shared between Db-1 and Rubezhnoe-1 (but not other accessions) are shown in orange. Two regions of haplotype sharing are visible: one in the *DUO3* promoter region, and another over the *DUO3* 3' UTR (note that in Araport11 the *DUO3* 3' UTR is incorrectly annotated as a separate gene). **(D–E)** Violinplots showing the gene expression of **(D)** *UBC20* and **(E)** *CYCB1;2* in sperm and vegetative nuclei separated by the haplotype of *PSV1*. Sperm nuclei that inherit the Db-1 haplotype of *PSV1* have significantly increased expression of *UBC20* and decreased expression of *CYCB1;2*, compared to sister nuclei that inherit the Col-0 haplotype. The data underlying this figure can be found in datasets 4 and 5 at https://doi.org/10.5281/zenodo.14864053.

## Identification of cell-type specific *cis*- and *trans*-eQTLs

Since the Col-0 × Db-1 dataset yielded such improved vegetative nucleus recovery, we extended our eQTL mapping to model cell-type-specific eQTLs. Cell-type clusters were predicted using a Gaussian mixture model trained on principal components of gene expression, and interaction terms of cell-type with haplotype were used in place of haplotype estimates in the model. Using this approach, we recovered a total of 1,516 eQTLs, including 550 (36.3%) *cis*- and 966 (63.7%) *trans*- acting eQTLs (Fig 6D). The vast majority of eQTLs were specific to only one cell-type, and only 17 *cis*-eQTLs and 2 *trans*-eQTLs were detectable in both sperm and vegetative nuclei. The lack of shared *cis*-eQTLs between cell-types is explained by the very different transcriptomes of sperm and vegetative nuclei – genes with *cis*-eQTLs are more highly expressed on average in the cell type where the eQTL is detected (S13 Fig). Approximately 87.4% of *cis*-eQTLs were specific to sperm nuclei. In contrast, 98.9% of the *trans*-eQTLs overlapping the *PSV1* locus were specific to the sperm (Fig 6D), providing further evidence that the effects of *PSV1* are restricted to sperm nuclei. Interestingly, of the *trans-eQTL* hits mapping to the *CPV1* locus in the Chromosome 1 centromere, 13.6% were detectable in vegetative nuclei, compared to 87.1% in sperm nuclei (Fig 6D). For example, gene expression of the pectin lyase gene *PLL1* (AT1G14420) was associated with the *CPV1* locus in vegetative nuclei – nuclei that inherit the Db-1 haplotype of *CPV1* have higher expression of *PLL1* in vegetative nuclei than sister nuclei that inherit the Col-0 haplotype (S14 Fig). This suggests either that *CPV1* affects different sets of genes in sperm and vegetative nuclei, or possibly that there are two independent variants underlying *CPV1* which affect sperm and vegetative gene expression, respectively.

## Fine mapping of the *PSV1* locus implicates the mitotic cell cycle regulator *DUO3*

In order to determine what the gene underlying *PSV1* might be, we performed fine mapping of the *PSV1* locus using PCA. This procedure borrows power across many genes with correlated variation in expression, in order to produce more stable "latent" phenotypes for eQTL mapping [38]. Since the effect of *PSV1* on gene expression is restricted to sperm nuclei, we first filtered the Col-0 × Db-1 expression matrix to retain only barcodes predicted to represent high-confidence sperm nuclei with high sequencing depth, resulting in 3,331 high-quality sperm nuclei. We then performed PCA on the expression of 398 genes which had a sperm-nucleus specific *trans*-eQTL mapping to *PSV1* (and no *cis*-eQTLs), and used these principal components as phenotypes for QTL mapping. We found that principal component 1 was extremely strongly correlated with *PSV1* inheritance, with a QTL peak falling in the 25 kb haplotype bin from 26,175 kb to 26,200 kb (Fig 7A). Using a LOD drop of 1.5 to estimate confidence intervals, the causal locus was predicted to fall within the 75 kb interval 26,150 kb to 26,225 kb, containing 20 protein-coding genes. One of the six genes located in the central 25 kb peak was the homeodomain-like transcription factor *DUO3* (Fig 7A, inset), which has previously been shown to be required for mitotic cell-cycle progression and sperm cell differentiation during male germline development [39]. Specifically, *duo3* mutants have delayed cell-cycle progression during pollen mitosis II, with generative nuclei arresting at the G2 to M phase transition, resulting in most pollen remaining bicellular at anthesis [39].

We compared the assemblies of the six parental accessions, to determine if there was genomic variation at or near the *DUO3* locus that could explain the *PSV1* eQTLs [15]. Across the 6 accessions, 9 non-synonymous mutations were identified in the coding sequence of *DUO3*, but none of these were distributed between the accessions in a pattern that could

explain the altered gene expression changes in individuals that specifically inherit the Db-1 or Rubezhnoe-1 allele of *PSV1* (Fig 7B). The *DUO3* promoter also contains a complex, tandem repeat containing region, approximately 1.5 kb upstream of the annotated start codon of *DUO3,* which is expressed in pollen as a predicted long non-coding RNA (AT1G64563). In the forthcoming TAIR12 reference assembly of Col-0 (T. Berardini, TAIR, personal communication), this region contains approximately 24.5 copies of a 32 nt tandem repeat (S15 Fig). The repeat region contains significant copy number variation between the different accessions (Fig 7B), with all 6 accessions carrying a different haplotype – for example, Db-1 has 34.5 copies, whilst Rubezhnoe-1 has 31.5 (S15 Fig). Variation in tandem repeats has been shown to have an effect on chromatin state, and may alter the expression of the adjacent *DUO3* gene between the accessions [40–43]. We also saw haplotype sharing between Db-1 and Rubezhnoe-1 in two locations close to the *DUO3* locus: in a promoter region approximately 1.2 kb upstream of the *DUO3* start codon, and over the *DUO3* 3' UTR (Fig 7C). These variants could potentially have an impact on *DUO3* gene expression. Despite this, however, we did not detect any *cis*-eQTLs in *DUO3* in either dataset. *DUO3* is expressed at a low level however, and is only detectable in 14.1% of Col-0 × Db-1 nuclei, meaning that there may not be sufficient power for eQTL mapping.

We speculated that if *DUO3* were the causal gene explaining *PSV1,* then *PSV1*-regulated genes might be expected to be enriched for cell-cycle regulators. Using a curated list of known cell-cycle factors [44], we found a significant enrichment of cell-cycle genes amongst genes with a *PSV1 trans*-eQTL (hypergeometric test $p = 0.032$). In particular, we identified a number of regulators known to affect the G2 to M phase transition – for example, one of the most significant *trans*-eQTL hits identified at *PSV1* was for the ubiquitin E2-ligase *UBC20* (Fig 6B), which is a component of the Anaphase promoting complex expressed during G2 to M phase transition, that promotes entry into mitosis [45,46]. Nuclei that inherit the Db-1 haplotype of *PSV1* have almost 2-fold higher expression of *UBC20* than do sister nuclei that inherit the Col-0 haplotype (Fig 7D). We also saw *trans*-eQTL hits at *PSV1* in the B1-type cyclin *CYCB1;2* (Fig 7E), the B1-type cyclin associated kinase *CDKB2;2*, and the B2-type cyclin *CYCB2;3*. B-type cyclins are also expressed during G2 to M phase transition and are required for the proper timing of mitosis [46,47]. Interestingly, the B1-type cyclin *CYCB1;2* has decreased expression in the nuclei that inherit the Db-1 haplotype of *PSV1*, whereas the expression of *CDKB2;2* and *CYCB2;3* is increased. The A1-type cyclin *CYCA1;1*, which also accumulates during the G2 to M phase transition [46], and the D-type cyclins which control G1 to S phase transition, were not expressed sufficiently enough to be tested. *CYCB1;1*, whose expression was previously shown to be unaffected by *duo3* mutation [39], was also not expressed sufficiently in mature pollen to perform eQTL mapping.

Another gene required for sperm cell specification is the R2R3 Myb transcription factor *DUO1* [20,21]. Similarly to *duo3* mutants, *duo1* mutants arrest at the bicellular stage of pollen development, and fail to enter mitosis. *DUO1* and *DUO3* were previously shown to have overlapping but distinct targets [20,21,39]. We identified a *trans*-eQTL peak at *PSV1* affecting the expression of *DUO1* (S16A Fig), indicating that *PSV1* might act upstream of *DUO1.* Consistent with this, we also saw a number of known *DUO1* targets amongst genes with *PSV1 trans*-eQTLs, including *MGH3*, *DAZ3*, *DAN1*, and *MAPKKK20* (S16B Fig).

In summary, we were able to reliably detect a strong *trans*-eQTL hotspot, *PSV1*, affecting the expression of many genes in pollen sperm nuclei across several datasets. Fine mapping of this locus implicated the male-germline cell-cycle regulator *DUO3* as a strong candidate for the causal locus. Analysis of the genes affected by *PSV1* indicate that it is likely to encode a major regulator of cell-cycle progression and sperm-nucleus specification acting upstream of *DUO1*. Although these findings are consistent with *DUO3* as the causal gene, the causal variant that explains *PSV1* remains elusive.

## Discussion

Here, we report the use of snRNA-seq to infer meiotic recombination patterns in individual Arabidopsis pollen nuclei. Patterns of recombination match to expected landscapes previously assessed by standard whole genome resequencing of F2 populations [28]. We demonstrate that the resolution of the inferred haplotypes is sufficiently high to map eQTLs

using correlations between haplotype assortment and gene expression variation. The major benefit of this approach is that eQTLs can be mapped using a small number of founder individuals in a single generation, thereby accelerating the process by which we can understand gene interactions and regulatory dynamics.

**Single nucleus eQTL mapping identifies a potential master regulator of pollen sperm development**

We mapped a major *trans*-eQTL hotspot in the Db-1 and Rubezhnoe-1 accessions to a location on Chromosome 1, which we named *PSV1*. The position of *PSV1* is coincident with the location of a known cell-cycle regulator called *DUO3*, that controls pollen sperm nucleus development [39]. Mutations in *DUO3* cause extreme slow-down in cell cycle progression, specifically at the G2 stage of pollen mitosis II, when the generative nucleus undergoes symmetric division to produce two sperm nuclei [39]. We found that variation in *PSV1* is associated with altered expression of many genes that are exclusive to sperm nuclei, including key factors such as the histone 3.3 variant *MGH3* [22], and the poly(A) binding proteins *PAB6* and *PAB7* [37,48]. Several B-type cyclins, that control the transition from G2 to M phase, were also affected in their expression, as well as members of the anaphase-promoting complex [45]. Finally, we found that the expression of a key Myb transcription factor controlling pollen mitosis II, *DUO1* [21], as well as many known DUO1 targets [20], had expression differences that correlated with the inheritance of *PSV1*. This indicates that *PSV1* encodes a transcription factor acting upstream of *DUO1* in controlling pollen nucleus division and maturation, heavily implicating *DUO3* as the causal locus.

Although the causal variation at the *PSV1* locus could not yet be fully resolved, we found that there is hyperallelic variation in a tandem repeat upstream of the *DUO3* gene. Tandem repeat copy number variation is known to contribute to variation in both molecular and physiological phenotypes, by affecting both chromatin structure and gene expression regulation [40–43]. Copy number variation at short tandem repeats, which has been difficult to identify using short read sequencing methods, is thought to explain some of the "missing heritability" of polygenic traits in humans and other organisms [49].

Interestingly, unlike mutations in *DUO1, duo3* mutations do not completely block entry of the generative nucleus into mitosis [21,39]. Instead, mitosis is delayed so that only 20% of pollen have divided at the time of anthesis. This indicates that DUO3 plays a role in coordinating the timing of pollen development, which must be carefully synchronized to achieve successful fertilization [50]. Natural variation of *DUO3* expression or function could therefore be a means to vary the time required for pollen maturation. Indeed, the differences in expression of cell-cycle related genes in pollen that inherit the Db-1 allele of *PSV1*, may be caused by enhanced or delayed developmental timing in these nuclei. It is well established that the state of pollen development at the moment of dehiscence varies between species, with many, like Arabidopsis and other *Brassicaceae*, releasing tricellular pollen, and others, like *Solanaceae*, releasing bicellular pollen that only undergoes mitosis II after pollination, within the migrating pollen tube [51]. Phylogenetic evidence indicates that switches between releasing bicellular and tricellular pollen have occurred multiple times in many different angiosperm lineages [51], however the molecular mechanisms underpinning these switches are not yet clear.

DUO3 is also involved in the maturation of sperm after pollen mitosis II, since genetic analysis demonstrated it is required for the expression of sperm nucleus specific genes [39]. It is possible that DUO3 performs this role by continuing to regulate cell-cycle speed after the formation of tricellular pollen, and could explain why we detect gene expression differences that correlate with the haplotype of the *DUO3* locus, even in mature sperm nuclei. The cell cycle phase of mature pollen is still surprisingly controversial, with some studies suggesting that Arabidopsis pollen is greater than 1C [52,53], i.e., has already progressed into S phase, and others indicating that pausing occurs in the G1 phase whilst the pollen is still 1C [54,55]. Furthermore, the extent of variation of pollen maturation timing within species has not yet been examined, and it is not even clear to what extent pollen development is synchronized between gametes in the same anther or flower [50]. Our results indicate that there is likely to be natural variation in regulatory networks that have broad consequences for sperm cell specification and pollen maturation. Further work is therefore required to understand the impact of this variation on cellular, temporal and physiological levels.

## The challenges and benefits of haplotyping pollen with snRNA-seq

Measuring gamete haplotypes using snRNA-seq has broad utility for genomics and fundamental molecular biology research. It has been demonstrated that genotypes created using gamete single cell sequencing can be used to disentangle complex genome assemblies and resolve the haplotypes of polyploid species such as potato [56,57]. Identification of recombination breakpoints in single gametes not only provides haplotype information for eQTL mapping, but also facilitates the study of recombination itself, through measurements of overall recombination rates or landscapes [58], or by identifying meiotic abnormalities such as unequal crossover events [56]. snRNA-seq also overcomes the disadvantages of pooled-sequencing approaches, where many different recombinant genomes are pooled in one library and recombination breakpoints are searched for using long- or linked-read sequencing [59]. The number of meiosis that can be sampled using snRNA-seq significantly outstrips that which can be done using traditional resequencing approaches of recombinant populations [28], providing much greater statistical power to downstream analyses. Despite this, the application of single cell sequencing technologies to recombination analysis is still in its infancy, meaning that methodologies and best-practices are yet to be fully established.

A major challenge of snRNA-seq is determining how batch effects across libraries influence gene expression measurements [9,60]. Several complex methods for normalization and integration of gene expression count matrices exist, which use alignment of similar cells and data warping to achieve integration [61,62]. However, it is unclear whether batch effects have an impact on the measurement of recombination or haplotyping, and how best these confounding factors can be mitigated. For example, varying rates of doublets or ambient nucleic acids [9,63,64] are likely to change the overall measured rate of recombination rate between samples, by introducing various degrees of background noise into markers, and artefactual barcodes into the dataset. To avoid this issue in our data, we performed snRNA-seq of pooled pollen from different F1 hybrids in the same 10x library, and used known SNPs to demultiplex the nuclei originating from different hybrids. In future, larger experiments may necessitate the dividing of samples across two or more libraries, however. Further study is therefore required to understand the nature of batch effects and identify methods to remove or control them.

Since existing methods for demultiplexing different genotypes in snRNA-seq were primarily designed for mixtures of homogeneous diploid cells, their suitability for heterogeneously recombined haploid cells remains unclear [10,11]. Recognizing this gap, we developed our own approach using an expectation maximization method to identify the most likely genotype of each nucleus. This method solves the challenges unique to demultiplexing heterogeneous gamete populations originating from different heterozygous hybrids. When pooling gametes from genetically different parents, care should be taken to ensure that the different genotypes are distinct enough to be distinguishable and correctly demultiplexed. Where pooling is not possible due to low genetic diversity, experiments must be carefully designed to mitigate batch effects and prevent confounding [10,11].

We modified the recently developed rHMM method to predict meiotic recombination patterns between the two parental genotypes, using SNPs and indels as markers [26]. The benefit of the rHMM method is that it uses sparse transitions to prevent over-segmentation of chromosomes which is not biologically plausible due to crossover interference [27]. However, these methods could be further improved by implementing architectures that can also directly learn the non-uniform patterns of recombination along chromosomes [28]. The inevitable bias caused by the use of a reference genome could also be addressed by implementing graph-genome solutions for recombination mapping [65]. Even more flexible solutions for genotyping and recombination mapping will be required in future: for crossing strategies of greater complexity; for the analysis of F2 samples from multi-parental crosses [28]; or for the genotyping of pollen from polyploid species with polysomic inheritance and double reduction patterns [57,66,67].

## The broader applications of QTL mapping using genetically heterogeneous cell mixtures

To perform eQTL mapping with single-cell sequencing data, a genetically heterogenous cell population is required. This was previously achieved for humans by mixing cell-lines derived from many genetically distinct individuals together [5–8].

As well as being laborious to produce, however, the diversity of such populations is limited by the number of input individuals. Here we show that by using gametes, we can exploit meiosis to produce snRNA-seq datasets where each individual cell or nucleus is genetically unique. This means that the statistical power to detect eQTLs is constrained only by the number of nuclei collected, and even *trans*-eQTLs with relatively small effect sizes can be detected. A disadvantage of this approach is that the resolution of eQTLs is limited in part by the rate of recombination and the process of crossover interference. To achieve single-gene resolution of eQTLS, one solution might be to collect gametes from many individuals of segregating or recombinant inbred populations that have been through several rounds of meiosis to reduce genetic linkage [12,13,68]. Alternatively, QTL resolution can also be improved by performing mapping experiments in mutant backgrounds with increased rates of recombination [69].

Previous studies have shown that snRNA-seq is capable of capturing the full complexity of the pollen developmental pathway, from post-meiotic microspore nuclei to developmentally mature sperm and vegetative nuclei [19]. Future work could therefore identify eQTLs that are developmental stage-specific and shed light on how natural genetic variants modulate the developmental trajectory of pollen. eQTL mapping with snRNA-seq is also possible in diploid cells from segregating populations, provided sufficiently high numbers of individuals are used as input. This has been demonstrated in *Caenorhabditis elegans* [12] and *Saccharomyces cerevisiae* [13], opening up the possibility of mapping eQTLs using only single cell RNA-seq in cell-types other than haploid gametes. Furthermore, other single cell sequencing modalities, such as assay for accessible chromatin (ATAC) sequencing or DNA methylation sequencing [9] also provide molecular phenotypes which could then be linked to haplotype information. For example, bulk ATAC-seq has previously been used to identify chromatin accessibility QTLs [2], that could also be identifiable by single cell approaches. Multi-omics single cell modalities, that allow the analysis of whole genomes, and RNA or chromatin accessibility at the same time [70,71], could allow even more precise mapping of recombination breakpoints and potentially even identify the subtle footprints of meiotic gene conversions [72]. As new snRNA-seq technologies amplify the number of cells or nuclei that can be assayed in a single experiment [73,74], the sensitivity of QTL mapping approaches will also be greatly increased, making molecular QTL mapping with single cells an increasingly exciting method for understanding the genetics of complex biological processes.

## Methods

### Plant material and growth conditions

Plant material was generated by crossing Col-0 to five different accessions: Db-1, Kar-1, Rubezhnoe-1, Ms-0 and Tsu-0. Col-0 was used as female parent for Kar-1 and Ms-0, and male parent for Db-1, Rubezhnoe-1 and Tsu-0. Plants were grown in a Percival growth chamber under long day conditions for either up to 2 weeks at 25°C, or up to 4 weeks at 18°C, until they flowered. Pollen was collected from mature flowers just prior to anthesis [75], and stored at –80°C. For the first dataset with five hybrids, only pollen from plants grown at 25°C was used. For the second dataset with only Col-0 × Db-1 pollen, a mixture of pollen from plants grown at 25 and 18°C was used.

### Single-nucleus library preparation and sequencing

From each hybrid, 30 flowers were pooled in 2 ml pre-chilled scWPB buffer (0.2 M Tris-HCl, 4 mM MgCl$_2$*6H$_2$O, 2 mM EDTA*Na$_2$*2H$_2$O, 86 mM NaCl, 10 mM Na$_2$S$_2$O$_5$, 250 mM Sucrose, 1% PVP-10, 0.5 mM Spermine*4HCl and 0.5 mM Spermidine), which was modified from the original buffer composition [76]. Pollen solution was agitated by vortexing and nuclei were extracted using a modified bursting technique [77]. Initially a Celltrics 100 µm cell strainer was applied for prefiltering pollen, followed by 5 µm strainer for pollen collection and bursting. Extracted nuclei were stained with 1 µg/ml DAPI and sorted using a FACSAria Fusion using 0-32-0 mode and a 70 µl nozzle. For the samples used to generate the first and second datasets, approximately 53,000 and 55,500 events were sorted, respectively (S1 Fig, S10 Fig). Sorted nuclei were collected in 1x PBS collection buffer with 1.0% BSA and 0.2 U/µl Protector RNase Inhibitor. For the

sample used to generate the first dataset, sorted nuclei were centrifuged at 500 rcf for 5 min at 4°C. The supernatant was removed, and pelleted nuclei were resuspended in 43 μl of collection buffer. For the sample used to generate the second dataset, nuclei concentration was estimated using a Luna FX cell counter, and approximately 13,150 nuclei were loaded in a 43 μl volume onto the Chromium controller, without centrifugation. The Chromium Next GEM Single Cell 3′ Reagent Kits v3.1 (Dual Index) 10x Genomics kit was used for library preparation following the manufacturer's instructions. Sequencing of the five pooled hybrid library was performed by Novogene on an Illumina NovaSeq 6,000. Sequencing of the Col-0 × Db-1 library was performed by BGI using DNBSEQ technology.

## Preparation of reference genomes and annotations for snRNA-seq analysis

The TAIR12 community consensus (Col-CC) genome version 1 without 45S rDNA repeats (GCA_028009825.1) was used as a reference genome for Col-0. Chromosome-scale PacBio HiFi assemblies of the Db-1, Kar-1, Ms-0, Rubezhnoe-1 and Tsu-0 genomes [15] were aligned to the Col-0 assembly using minimap2 version 2.26, with the "asm20" preset [78]. Z-drop parameters were set to -z1000,100 to allow alignment through larger insertions. Gene annotations for all 6 parental assemblies were generated by coordinate-conversion of the Araport11 annotation [79] using LiftOff version 1.6.3 [80]. Synteny analysis was performed using syri version 1.6.5 [81] and msyd version 0.10 [82]. Syri was used to identify pairwise syntenic regions for each of the five Parent 2 genomes, compared to Col-0. Mysd was then used to identify core-syntenic regions that are common to all 6 accessions. For alignments used in genotyping analysis, biallelic SNPs and indels shorter than 50 nt located in regions that were core-syntenic in all 6 accessions were retained and a VCF containing the consensus allele (using majority voting) of the five different Parent 2 accessions was produced using bcftools version 1.17 [16,83]. For single-nucleus genotype calling, biallelic SNPs located in regions that were core-syntenic in all 6 accessions were retained. For recombination mapping, SNPs and indels located in syntenic regions in each accession compared to Col-0, that also did not overlap any non-syntenic regions were retained, and "highly diverged regions" shorter than 5,000 nt were resolved to SNPs and indels where possible using semi-global alignment with parasail version 1.3.4 [84]. A VCF containing the alternate allele was produced for each accession for alignment using STAR diploid [16].

## Read alignment for genotyping analysis

Forward reads containing the cell barcode and unique molecular identifier (UMI) were trimmed to a length of 28 nt using SeqKit [85], to remove any non-barcode sequences. For genotyping analysis, reverse reads and corresponding barcodes were then mapped to the Col-0 reference using STARsolo version 2.7.11a [86,87] in consensus mode [16], with consensus variants from the five different parent 2 accessions. Cell barcodes were identified using the "1MM_multi" method. A maximum intron size of 20,000 nt was used. Multimapping reads or alignments with non-canonical splice junctions were filtered. The full list of parameters used is available in the pipeline scripts.

## Genotyping analysis

UMIs identifying SNPs from all genotypes were counted for each nucleus barcode using cellsnp-lite version 1.2.3 [88]. Since existing tools for demultiplexing single cell sequencing data using genetic variants generally expect subpopulations of the same class to be homogeneous diploid cells [10,11], we decided to develop a specialized approach for classifying heterogeneous recombinant haploid nuclei. For each cell/nucleus barcode, we identified positions where the majority of unique fragments supported the alternative allele. Positions where the majority of unique fragments supported the reference allele were discarded, since they could be explained by that position being inherited from Col-0. Starting with equal priors for all genotypes, we then used fractional assignment and expectation maximization to update the likelihood of observed SNP patterns being explained by each of the five Parent 2 genotypes, until the probabilities converged (to a total change in probability of <0.01) or a maximum of 1,000 iterations was reached. To estimate confidence intervals for these probabilities, 25 bootstraps were performed for each barcode with SNPs randomly subsampled with replacement. The genotype with

the highest mean probability across bootstraps was taken as the genotype assignment, and the average probability for that genotype was transformed using Phred-like scaling ($-\log_{10}$ (1 − probability)) to give a genotyping confidence score.

### Read alignment for recombination analysis

To maximize the number of possible markers for mapping of meiotic recombination, we took advantage of the ability of STAR spliced alignment software to map reads to two haplotypes at once (STAR diploid), by providing a VCF file containing SNPs and indels during genome indexing [16]. Since STAR diploid can only map to two haplotypes at a time, we used the genotype assignments to demultiplex mapped and unmapped reads from each cell barcode into five different FASTQ files representing the five different Parent 2 genotypes, using SAMtools version 1.17 [83]. Reads were then realigned using STAR diploid [16,86,87] with the same general alignment parameters as described in the section "Read alignment for genotyping analysis". After alignment to both haplotypes, STAR diploid projects alignment coordinates of reads mapping to the alternative haplotype into reference coordinates, and marks reads according to which (if either) of the two haplotypes they aligned better to.

### Recombination analysis

Multimapping reads, reads with unannotated SNPs or errors, and reads that aligned equally well to both haplotypes were filtered using SAMtools version 1.17 [83], to yield a set of high-confidence marker reads. Marker reads were grouped and deduplicated with UMIs corrected using the 1MM directional method [89], and represented positionally by their leftmost aligned base. The STAR diploid haplotype tag was used to infer which haplotype each UMI group supported, and any UMI groups that did not agree on supported haplotype were discarded. Alignment positions were grouped into non-overlapping 25 kb bins for recombination inference. To reduce the noise introduced into markers by ambient RNA, we used a method similar to SoupX [64]: marker distributions identified across all cell barcodes were used to create an estimate of background signal profile. Then, for each barcode, we used a 1 Mb sliding window to determine which of the two haplotypes was likely foreground and background at each genomic interval, and used the number of background markers to estimate the level of ambient DNA contamination of each barcode. The background contamination of each barcode was calculated by multiplying the predicted contamination level by the background profile, and subtracting this from the markers. We also masked any bins where there was a greater than 9:1 imbalance in the total number of markers supporting each haplotype across all nuclei (resulting from extreme allele specific expression), since ambient RNA mapping in these bins was found to particularly distort haplotype calling.

The rigid hidden Markov model was constructed as previously described [26] with a number of modifications. Since the pollen nuclei are haploid, the heterozygous state of the model was removed, and the two haplotype states were modelled as a pair of symmetric Poisson distributions (one Poisson modelling foreground and background counts, respectively). Since the chromosome sequence was modelled using equal length 25 kb bins, we chose a rigidity of 100 for the model, approximately reflecting crossover interference in Arabidopsis that largely suppresses crossovers occurring <2.5 Mb apart [27]. However, since meiotic interference does not prevent crossovers close to the ends of chromosomes, we introduced a new parameter called terminal rigidity, which allows the model to enter and exit the sparse state cycles partway through a cycle at the ends of the chromosome. Terminal rigidity was set to 4, meaning individual paths through the model have a minimum terminal segment size of 100 kb. Transition states between haplotypes were initialized to approximate an overall recombination rate of 4.5 cM/Mb [28]. The rHMM code was executed in Python version 3.10.13 [90,91] using the pomegranate library version 0.14.8 for probabilistic modelling [92].

### Cell barcode whitelisting

To perform whitelisting of cell barcodes, the number of effective markers, the genotyping confidence score, and several metrics derived from the rHMM emission scores were used. The HMM score was defined as the simple accuracy score

of the HMM predictions against the input markers, transformed using Phred-like scaling. The HMM uncertainty score was defined by taking the absolute difference between the discretized haplotype prediction and the haplotype probability score at each position, then calculating the $\log_{10}$ transformed area under the curve. Cell barcodes with low confidence genotyping assignments or rHMM emission scores are likely to have resulted from technical artefacts such as doublets, high levels of ambient RNA, or very sparse coverage. For the first dataset, which had genotyping scores, a threshold of examples with genotyping probability above the 25th percentile for their assigned genotype was used to produce initial labels to fit a random forest model to four metrics, to separate high-confidence nuclei from artefactual examples. Finally, doublets containing both sperm and vegetative nuclei were identified from expression data using a method similar to DoubletFinder [63]. Initial clusters from principal components were used to generate synthetic doublets and these were transformed to principal component space. K nearest neighbors classification was then used to identify barcodes whose expression profiles were closer to synthetic doublets than to other genuine cell barcodes. These barcodes were removed. This resulted in a final set of 1,394 nuclei which were taken for downstream analysis. For the second dataset, which only contained a single parental genotype and so did not have genotyping scores, initial labels were taken by thresholding using an HMM score >4 or HMM uncertainty <2.5, and a $\log_{10}$ minimum of 2.5 markers. These initial labels were then used to fit a multivariate gaussian mixture model on HMM score and HMM uncertainty to separate high-confidence nuclei from artefactual examples. Doublet filtering from expression data was performed as for the first dataset.

### Quantification of gene expression

Count matrices for whitelisted cell barcodes were calculated from BAM files generated by STARsolo and STAR diploid. UMIs were deduplicated using the 1MM directional method [89]. Matrices were also filtered to remove genes that were expressed in fewer than 10 unique cell barcodes. Nuclei expression profiles were normalized by sequencing depth and $\log_2$ transformed. These matrices were used to perform principal components and marker gene expression analyses.

For analyses of publicly available snRNA-seq of Col-0 pollen at different developmental stages, count matrices, cluster labels and UMAP projections were downloaded from NCBI GEO accession GSE202422 [19]. UMAP plots were generated using matplotlib [93] and seaborn [94].

### eQTL mapping analysis

To perform eQTL mapping analysis, the gene expression matrices described in the section "Single-nucleus expression analysis" were further filtered to retain only genes that were expressed at detectable levels (i.e., with one or more unique read) in at least 5% of nuclei. All nuclei that passed quality filtering described in the section "Cell barcode whitelisting" were retained for eQTL analyses, including vegetative nuclei. We used ordinary least squares linear regression to model the relationship between the haplotype in 25 kb bins and the expression of each gene, whilst controlling for the parental genotype of each nucleus and the first two principal components. For a given gene $x$ and haplotype bin $p$, the formula used for modelling was:

$$Exprs_x \sim Haplo_{pn} + \ldots + Haplo_{pN} + Geno_n + \ldots + Geno_{N-1} + PC_i + \ldots + PC_M$$

where $Exprs_x$ represents the $\log_2$ transformed expression of gene $x$, $N$ represents the total number of parent 2 genotypes, and $Haplo_{pn} + \ldots + Haplo_{pN}$ encodes the haplotype of position $p$ for each parent 2 genotype $n$ – this is set to zero if a nucleus is not of genotype $n$, or a floating point value between zero (encoding the Col-0 haplotype) and one (encoding the parent 2 haplotype) if it is. $Geno_n + \ldots + Geno_{N-1}$ represent dummy variables encoding the parent 2 genotype of nuclei for each parent 2 genotype $n$. $PC_i + \ldots + PC_M$ represents each $i$th principal component of $M$ total principal components. For the first dataset, with nuclei from 5 different F1 hybrid genotypes, the final formula contained four dummy variables

encoding genotype, five variables encoding haplotype, and two variables encoding the first two principal components. For the second dataset, which contained only nuclei from a single F1 hybrid genotype, the model simplifies to just one variable encoding haplotype, plus two variables encoding the principal components. Modelling was performed in Python version 3.10.13 [90,91] using the statsmodels package version 0.14.0 for statistical modelling [95].

To assess the overall significance of eQTLs across all haplotypes, a log ratio test (LRT) was performed by modelling the expression of each gene using a nested model without haplotype variables, and comparing this to the full model. LRT statistics were used to calculate LOD scores. To correct for multiple testing, we estimated the effective number of independent haplotypes using the Li and Ji eigenvalues of a correlation matrix method, and used this as the effective number of tests for Bonferroni correction [96]. Genes with at least one haplotype block that had an LRT FDR < 0.05 were considered significant. Multiple testing corrected $p$ values for each haplotype were also collected. Statistical modelling was performed in Python using the scientific Python stack and the statsmodels package [90,91,95].

To perform cell-type specific eQTL analysis, cell-types were estimated from PCA-transformed gene expression matrices using a two-component Gaussian-mixture model. Cell-type probability scores were multiplied by haplotype predictions to create a cell-type × haplotype interaction component, which was used in place of haplotype variables in the full model and LRTs, whilst also controlling for cell-type. The formula used for modelling was therefore:

$$Exprs_x \sim CT_t + \ldots + CT_{T-1} + CT.Haplo_{pt} + \ldots + CT.Haplo_{pT} + PC_i + \ldots + PC_M$$

where $CT_t + \ldots + CT_{T-1}$ are dummy variables encoding the cell-type of each nucleus for cell-type $t$, with $T$ representing the total number of identified cell types. $CT.Haplo_{pt} + \ldots + CT.Haplo_{pT}$ represent the interaction components between the haplotype at position $p$ and each cell-type $t$. Since the second dataset contained only nuclei derived from a Col-0 × Db-1 F1 hybrid, it was not necessary to explicitly model parental genotype for this dataset.

To identify significant eQTL peaks for each gene, a peak calling approach was developed using the LOD scores and $-\log_{10}$ FDR values. Peaks above a LOD score threshold of 3 with FDR < 0.05, with a prominence of at least 25% of the highest peak for each gene, and a minimum inter-peak distance of 3 Mb were retained. A LOD-drop of 1.5 was used to identify confidence intervals for each peak. eQTLs were categorized as *cis*-eQTLs when the confidence interval was within 2 Mb of the gene whose expression the haplotype correlated with. All other eQTLs were considered to be *trans*-eQTLs.

### *Trans*-eQTL hotspot analysis

To fine-map the *PSV1 trans*-eQTL hotspot, the Col-0 × Db-1 (experiment 2) gene expression matrix was filtered to retain only nuclei that were predicted with >95% confidence to not be doublets or artefactual, with >95% confidence to be sperm nuclei, and which had at least >2.5 $\log_{10}$ unique marker fragments used for recombination analysis. This resulted in 3,331 high quality nuclei being retained. The matrix was then filtered in the other dimension, to retain only genes that had a *trans*-eQTL overlapping the *PSV1* locus at Chr1: 26.2 Mb, also removing any genes that had a *cis*-eQTL. This resulted in 398 genes being retained. This filtered matrix was then used to perform principal components analysis with 5 components. Each principal component was tested against all haplotypes by comparing the full model with all components, to a nested model where the tested component was excluded, using an LRT. Principal component 1 was found to be very strongly correlated with *PSV1* inheritance. The 1.5 LOD drop method was used to estimate confidence intervals on position of the *PSV1* locus.

### Synteny analysis of the DUO3 locus

Dotplots of the *DUO3* locus and upstream region were generated using BLASTN and matplotlib [93,97]. SNPs in the *DUO3* locus identified using minimap2 [78] and syri [81] were visualized using IGV [98].

## Workflow management

All pipelines were written and executed using Snakemake [99], with Python scripts and the Scientific Python stack [90–95,100–102], as well as analysis using Jupyter notebooks and the Jupyterlab environment [103]. Figures were created using BioRender and Inkscape.

## Supporting information

**S1 Table. Sequencing summary statistics for the two snRNA-seq datasets.**
(XLSX)

**S1 Fig. Scatter plots showing the gating strategy of the fluorescence activated cell-sorting used for pollen nuclei from 5 F1 hybrids prior to single nucleus sequencing with the 10x platform. The black outlined quadrilaterals show the bounding regions within which events were sorted and captured.**
(TIFF)

**S2 Fig. Log-log scale regression plot showing the relationship between per-nucleus sequencing depth and the number of detected genes.** For a given sequencing depth, more genes are detected per vegetative nucleus than per sperm nucleus, demonstrating the greater transcriptomic diversity of vegetative nuclei. The data underlying this figure can be found in dataset 1 at https://doi.org/10.5281/zenodo.14864053.
(TIFF)

**S3 Fig. Upset plot showing the SNP sharing patterns of the five parent 2 accessions. Db-1 and Tsu-0 have the most unique SNPs, explaining why genotyping confidence scores generated using expectation maximization are higher for these accessions.**
(TIFF)

**S4 Fig. (A)** Histogram showing the median distance between 25 kb genomic bins that contain at least one or more informative reads for each nucleus barcode. **(B)** Histograms showing the 95% confidence intervals of the positions of crossovers predicted by the rigid hidden Markov model. Crossovers called in chromosome arms (in blue) were mapped with much greater resolution than crossovers that were close to the centromere (in orange). The data underlying this figure can be found in dataset 2 at https://doi.org/10.5281/zenodo.14864053.
(TIFF)

**S5 Fig. Heatmap showing the inferred haplotypes of the 1,394 pollen nuclei in the dataset, ordered by total marker read coverage (from most to least well covered).** The data underlying this figure can be found in dataset 2 at https://doi.org/10.5281/zenodo.14864053.
(TIFF)

**S6 Fig. Recombination landscapes of the five F1 hybrids. Y axis shows the estimated recombination rate in centiMorgans per Megabase (cM/Mb).** Shaded areas represent the 95% confidence intervals of the cM/Mb estimates generated using 100 bootstrapped resamples of the nuclei. The data underlying this figure can be found in dataset 2 at https://doi.org/10.5281/zenodo.14864053.
(TIFF)

**S7 Fig. (A)** eQTL plot showing the haplotypes whose inheritance correlates with the expression of the MADS-box transcription factor AT3G12510. eQTL peaks are shown as vertical dashed lines with 1.5 LOD drop confidence intervals shown as grey shaded regions. The location of the AT3G12510 gene is shown as a solid vertical black line. The black line labelled "All" shows the FDR calculated from the log ratio test of all 5 parent 2 haplotypes compared to Col-0.

**(B)** Violinplot showing the gene expression of AT3G12510 in nuclei separated by the *cis*-haplotype (i.e., the haplotype at AT3G12510). Nuclei that inherit the Kar-1 haplotype of AT3G12510 have significantly reduced expression of AT3G12510, compared to sister nuclei that inherit the Col-0 haplotype. **(C)** The AT3G12510 gene model, showing major insertions compared to Col-0. In Kar-1, a LINE1 retrotransposable element is inserted into the AT3G12510 promoter, likely explaining the reduced expression of the AT3G12510 gene. The data underlying this figure can be found in datasets 1, 2 and 3 at https://doi.org/10.5281/zenodo.14864053.
(TIFF)

**S8 Fig.** **(A)** eQTL plot showing the haplotypes whose inheritance correlates with the expression of the poly(A) binding protein gene *PAB6* (AT3G16380). eQTL peaks are shown as vertical dashed lines with 1.5 LOD drop confidence intervals shown as grey shaded regions. The location of the *PAB6* gene is shown as a solid vertical black line. The black line labelled "All" shows the FDR calculated from the log ratio test of all 5 parent 2 haplotypes compared to Col-0. **(B)** Violinplot showing the gene expression of *PAB6* in nuclei separated by the haplotype of *PSV1*. Nuclei that inherit the Db-1 or Rubezhnoe-1 haplotype of *PSV1* have significantly increased expression of *PAB6*, compared to sister nuclei that inherit the Col-0 haplotype. **(C)** UMAP projection from Ichino and colleagues 2022, showing the expression of *PAB6* throughout the developmental stages of the pollen. *PAB6* is only expressed in the sperm nucleus cluster, and is absent from microspore, generative and vegetative nuclei, as well as from the soma. The data underlying this figure can be found in datasets 1, 2 and 3 at https://doi.org/10.5281/zenodo.14864053.
(TIFF)

**S9 Fig.** **(A)** Violinplot showing the average expression of genes with an eQTL peak at the *PSV1* locus in sperm and vegetative nuclei. Genes affected by *PSV1* tend to have greater expression in the sperm nuclei. **(B)** Violinplot showing the average expression of genes with an eQTL peak at the *PSV1* locus in various immature and mature pollen cell-types, identified in a snRNA-seq dataset from Ichino and colleagues 2022. **(C)** Venn-diagram showing the overlap of *PSV1 trans*-eQTLs that are significant in all genotypes combined, versus in Db-1 and Rubezhnoe-1 specifically. The data underlying this figure can be found in datasets 1 and 3 at https://doi.org/10.5281/zenodo.14864053.
(TIFF)

**S10 Fig. Scatter plots showing the gating strategy of the fluorescence activated cell-sorting for the Col-0 × Db-1 pollen nuclei used prior to single nucleus sequencing with the 10x platform. The black outlined quadrilaterals show the bounding regions within which events were sorted and captured.**
(TIFF)

**S11 Fig** **(A)** Scatter plot with marginal kernel density estimates, showing the first two principal components of the expression data for 7,458 Col-0 × Db-1 pollen nuclei. Nuclei form two unequally sized clusters, mostly separated by the first principal component. **(B)** Log-log scale regression plot showing the relationship between per-nucleus sequencing depth and the number of detected genes, for the Col-0 × Db-1 dataset. For a given sequencing depth, more genes are detected per vegetative nucleus than per sperm nucleus, demonstrating the greater transcriptomic diversity of vegetative nuclei. The data underlying this figure can be found in dataset 4 at https://doi.org/10.5281/zenodo.14864053.
(TIFF)

**S12 Fig.** **(A)** Histogram showing the absolute distance from identified eQTL peaks, to the gene whose expression they correlate with, for peaks identified as being likely to be caused by cis-variants. **(B)** Venn diagram showing the overlap of Db-1 versus Col-0 *cis*-eQTLs identified in the two datasets. The data underlying this figure can be found in dataset 5 at https://doi.org/10.5281/zenodo.14864053.
(TIFF)

**S13 Fig. Histogram showing the fold change in expression of genes with cell-type specific *cis*-eQTLs (detected in the Col-0 × Db-1 dataset), in sperm nuclei compared to vegetative nuclei.** The majority of genes are only expressed in the cell-type where the eQTL is detected. The data underlying this figure can be found in dataset 5 at https://doi.org/10.5281/zenodo.14864053.
(TIFF)

**S14 Fig.** **(A)** eQTL plot showing the haplotypes whose inheritance correlates with the expression of the pectin lyase *PLL1* (AT1G14420). eQTL peaks are shown as vertical dashed lines with 1.5 LOD drop confidence intervals shown as grey shaded regions. The location of the *PLL1* gene is shown as a solid vertical black line. The black line labelled "All" shows the FDR calculated from the log ratio test of all nucleus-types combined. **(B)** Violinplots showing the gene expression of *PLL1* in sperm and vegetative nuclei separated by the haplotype of *CPV1* (Chr1: 16.5 Mb). Vegetative nuclei that inherit the Db-1 haplotype of *CPV1* have higher expression of *PLL1* than sister nuclei that inherit the Col-0 haplotype. **(C)** UMAP projection from Ichino and colleagues 2022, showing the expression of *PLL1* throughout the developmental stages of the pollen. *PLL1* is expressed in the mature vegetative nucleus cluster, and is absent from microspore, generative and sperm nuclei. The data underlying this figure can be found in datasets 4 and 5 at https://doi.org/10.5281/zenodo.14864053.
(TIFF)

**S15 Fig. Bar plot showing the number of identified tandem repeats located in the *DUO3* promoter, for the genomes of different Arabidopsis accessions used in this study.**
(TIFF)

**S16 Fig. eQTL plot showing the haplotypes whose inheritance correlates with the expression of the Myb-transcription factor *DUO1* (AT3G60460) .** **(A)** eQTL peaks are shown as vertical dashed lines with 1.5 LOD drop confidence intervals shown as grey shaded regions. The location of the *DUO1* gene is shown as a solid vertical black line. The black line labelled "All" shows the FDR calculated from the log ratio test of all nucleus-types combined. **(B)** Violinplots showing the gene expression of various characterized *DUO1* targets in sperm and vegetative nuclei separated by the haplotype of *PSV1*. The data underlying this figure can be found in datasets 4 and 5 at https://doi.org/10.5281/zenodo.14864053.
(TIFF)

## Acknowledgements

We thank Craig Dent for helpful advice and comments on the manuscript.

## Author contributions

**Conceptualization:** Matthew T. Parker, Korbinian Schneeberger.

**Formal analysis:** Matthew T. Parker, Sergio Tusso.

**Methodology:** Matthew T. Parker, Samija Amar, José A. Campoy, Kristin Krause, Magdalena Marek, Bruno Huettel, Korbinian Schneeberger.

**Resources:** Samija Amar, José A. Campoy, Kristin Krause, Bruno Huettel.

**Visualization:** Matthew T. Parker.

**Writing – original draft:** Matthew T. Parker.

**Writing – review & editing:** Matthew T. Parker.

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
