## [Editor Report · Decision Letter 0]

17 Jan 2025

Dear Dr Schneeberger,

Thank you for submitting your revised manuscript from Review Commons entitled "Expression quantitative trait locus mapping in recombinant gametes using single nucleus RNA sequencing" for consideration as a Research Article by PLOS Biology. Please accept my sincere apologies for the delay in getting back to you with feedback as we consulted with an academic editor about your manuscript and rebuttal.

Your manuscript has now been evaluated by the PLOS Biology editorial staff, as well as by an academic editor with relevant expertise, and I am writing to let you know that we would like to send your revision back to the previous reviewers at Review Commons.

IMPORTANT: After discussions with the editorial team, we think that your manuscript would be a better fit as a 'Methods and Resources' article at the journal. During resubmission (see details below), we would be grateful if you could please tick 'Methods and Resources' as the article type in the dropdown menu in the online submission form.

Before we can send your manuscript for re-review, we need you to complete your submission by providing the metadata that is required for full assessment. To this end, please login to Editorial Manager where you will find the paper in the 'Submissions Needing Revisions' folder on your homepage. Please click 'Revise Submission' from the Action Links and complete all additional questions in the submission questionnaire.

Once your full submission is complete, your paper will undergo a series of checks in preparation for peer review. After your manuscript has passed the checks it will be sent out for review. To provide the metadata for your submission, please Login to Editorial Manager (https://www.editorialmanager.com/pbiology) within two working days, i.e. by Jan 19 2025 11:59PM.

Kind regards,

Richard

Richard Hodge, PhD

rhodge@plos.org

PLOS

---

## [Decision Letter · Decision Letter 1]

5 Feb 2025

Dear Dr Schneeberger,

Thank you for your patience while we considered your revised manuscript from Review Commons entitled "Expression quantitative trait locus mapping in recombinant gametes using single nucleus RNA sequencing" for publication as a Methods and Resources at PLOS Biology. This revised version of your manuscript has been evaluated by the PLOS Biology editors, the Academic Editor and the original reviewers at Review Commons.

Based on the reviews, I am pleased to say that we are likely to accept this manuscript for publication, provided you satisfactorily address the remaining point raised by the Reviewer #2. Please also make sure to address the following data and other policy-related requests that I have provided below (A-F):

(A) We routinely suggest changes to titles to ensure maximum accessibility for a broad, non-specialist readership. In this case, we would suggest a minor edit to the title, as follows. Please ensure you change both the manuscript file and the online submission system, as they need to match for final acceptance:

“Scalable eQTL mapping using single-nucleus RNA-sequencing of recombined gametes from a small number of individuals”

(B) You may be aware of the PLOS Data Policy, which requires that all data be made available without restriction: http://journals.plos.org/plosbiology/s/data-availability. For more information, please also see this editorial: http://dx.doi.org/10.1371/journal.pbio.1001797

-Supplementary files (e.g., excel). Please ensure that all data files are uploaded as 'Supporting Information' and are invariably referred to (in the manuscript, figure legends, and the Description field when uploading your files) using the following format verbatim: S1 Data, S2 Data, etc. Multiple panels of a single or even several figures can be included as multiple sheets in one excel file that is saved using exactly the following convention: S1_Data.xlsx (using an underscore).

-Deposition in a publicly available repository. Please also provide the accession code or a reviewer link so that we may view your data before publication.

Figure 1B-E, 2A-D, 3A-C, 4A-F, 5A-D, 6A-D, 7A-E, 1S1, 1S2, 1S3, 2S1, 2S2, 2S3, 4S1A-C, 5S1A-C, 5S2A-C, 6S1, 6S2A-B, 6S3A-B, 6S4, 6S5A-C, 7S1, 7S2A-B

(C) Thank you for providing the snRNA-seq data in the ENA database (PRJEB77115). However, it seems that this data is not currently publicly available? We ask that you please make the sequencing data publicly available at this stage before publication.

(D) Please also ensure that each of the relevant figure legends in your manuscript include information on *WHERE THE UNDERLYING DATA CAN BE FOUND*, and ensure your supplemental data file/s has a legend.

(E) Please ensure that your Data Statement in the submission system accurately describes where your data can be found and is in final format, as it will be published as written there.

(F) Per journal policy, if you have generated any custom code during the course of this investigation, please make it available without restrictions. Please ensure that the code is sufficiently well documented and reusable, and that your Data Statement in the Editorial Manager submission system accurately describes where your code can be found.

We expect to receive your revised manuscript within two weeks.

*Published Peer Review History*

*Press*

Best regards,

Richard

Richard Hodge, PhD

rhodge@plos.org

Reviewer remarks:

Reviewer #1: Thank you for providing a detailed response to my original reviewers. They have all be satisfactorily addressed. This will make a nice contribution that will surely affect how people think about merging single-cell genomics with QTL mapping.

Reviewer #2: The revised manuscript has improved significantly. The authors have addressed all my previous comments and questions thoroughly. The additional technical details enhance the manuscript's value, especially for colleagues interested in applying this methodology to pollen or other systems. I have no further concerns.

Minor Comment:

In Figures 1S1 and 6S1, the legend states, "Events which were sorted and captured are shown in green." However, in the histogram, the green appears to indicate a high particle count, which is somewhat unclear.

---

## [Editor Report · Decision Letter 2]

25 Feb 2025

Dear Dr Schneeberger,

On behalf of my colleagues and the Academic Editor, Wenbo Ma, I am pleased to say that we can accept your manuscript for publication, provided you address any remaining formatting and reporting issues. These will be detailed in an email you should receive within 2-3 business days from our colleagues in the journal operations team; no action is required from you until then. Please note that we will not be able to formally accept your manuscript and schedule it for publication until you have completed any requested changes.

PRESS

Best wishes, 

Richard

Richard Hodge, PhD

rhodge@plos.org

PLOS
